# Connexin 37 sequestering of activated-ERK in the cytoplasm promotes p27-mediated endothelial cell cycle arrest

Bipul R Acharya[1,3] , Jennifer S Fang[4], Erin D Jeffery[5] , Nicholas W Chavkin[1] , Gael Genet[1] , Hema Vasavada[2], Elizabeth A Nelson[1], Gloria M Sheynkman[5,6,7,8], Martin J Humphries[3], Karen K Hirschi[1,2]

Connexin37-mediated regulation of cell cycle modulators and, consequently, growth arrest lack mechanistic understanding. We previously showed that arterial shear stress up-regulates Cx37 in endothelial cells and activates a Notch/Cx37/p27 signaling axis to promote G1 cell cycle arrest, and this is required to enable arterial gene expression. However, how induced expression of a gap junction protein, Cx37, up-regulates cyclin-dependent kinase inhibitor p27 to enable endothelial growth suppression and arterial specification is unclear. Herein, we fill this knowledge gap by expressing wild-type and regulatory domain mutants of Cx37 in cultured endothelial cells expressing the Fucci cell cycle reporter. We determined that both the channel-forming and cytoplasmic tail domains of Cx37 are required for p27 up-regulation and late G1 arrest. Mechanistically, the cytoplasmic tail domain of Cx37 interacts with, and sequesters, activated ERK in the cytoplasm. This then stabilizes pERK nuclear target Foxo3a, which up-regulates p27 transcription. Consistent with previous studies, we found this Cx37/pERK/Foxo3a/p27 signaling axis functions downstream of arterial shear stress to promote endothelial late G1 state and enable up-regulation of arterial genes.

## Introduction

Multiple signaling pathways coordinately regulate endothelial cell (EC) proliferation and subtype specification during formation and remodeling of the blood circulatory system (1, 2). We, and others, have shown that fluid shear stress (FSS), at magnitudes typical of arteries and arterioles, activates a Notch–Connexin37 (Cx37)-p27Kip1 (p27) signaling axis to promote G1 cell cycle arrest of ECs, which is required to enable arterial gene expression (3, 4, 5). However, we lack the mechanistic understanding of how Cx37, a gap junction (GJ) protein, up-regulates p27 to promote G1 arrest in ECs to enable arterial specification.

Connexins are a family of 21 tetramembrane-spanning proteins, which form transport channels across membranes and between adjacent cells, called hemichannels and GJ channels, respectively (6). Intracellular trafficking targets connexin proteins from the trans-Golgi network to the cell membrane and guides their internalization for degradation, recycling, and signal transduction (7, 8). These events are thought to be regulated by the posttranslational modifications of the Cx protein C-terminal tail (9, 10, 11, 12), and Cx endocytosis is often modulated by microtubule (MT)-binding proteins on MT-filament tracks (13, 14). Connexins, and GJ channels that they comprise, regulate growth and differentiation of many cell types (15, 16, 17). Highly proliferative tumor cells exhibit reduced GJ numbers, suggesting a correlation between cell cycle suppression and GJ formation and/or function (18). The mechanisms are likely to be complex given that Cx can exhibit channel-dependent and -independent functions. For example, Cx43 interacts with Wnt signaling to control cell cycle regulators cyclin D1 and c-Myc to promote G1 arrest (16). Cx37 was earlier shown to suppress tumor cell growth in a manner requiring both the connexin C-terminus and channel-forming domain (19, 20, 21, 22), but how these individual domains interact with one another and downstream signaling effectors to regulate cell cycle progression is yet undefined. Furthermore, earlier Cx37 structure–function studies were performed in cancer cell lines, thus it remains unclear whether these findings hold true in vascular ECs. Lastly, although Cx37-mediated cell cycle regulation likely involves posttranslational modification of its C-terminus and subsequent changes to channel function (20, 21, 22, 23), downstream effects on

[1]Department of Cell Biology, Cardiovascular Research Center, University of Virginia School of Medicine, Charlottesville, VA, USA   [2]Departments of Medicine and Genetics, Cardiovascular Research Center, Yale University School of Medicine, New Haven, CT, USA   [3]Wellcome Centre for Cell-Matrix Research, Faculty of Biology, Medicine & Health, Manchester Academic Health Science Centre, University of Manchester, Manchester, UK   [4]Department of Molecular Biology & Biochemistry, University of California at Irvine, Irvine, CA, USA   [5]Department of Molecular Physiology and Biophysics, University of Virginia School of Medicine, Charlottesville, VA, USA   [6]Department of Biochemistry and Molecular Genetics, University of Virginia, Charlottesville, VA, USA   [7]Center for Public Health Genomics, University of Virginia, Charlottesville, VA, USA   [8]UVA Comprehensive Cancer Center, University of Virginia, Charlottesville, VA, USA

Correspondence: bipul.acharya@manchester.ac.uk; kkh4yy@virginia.edu

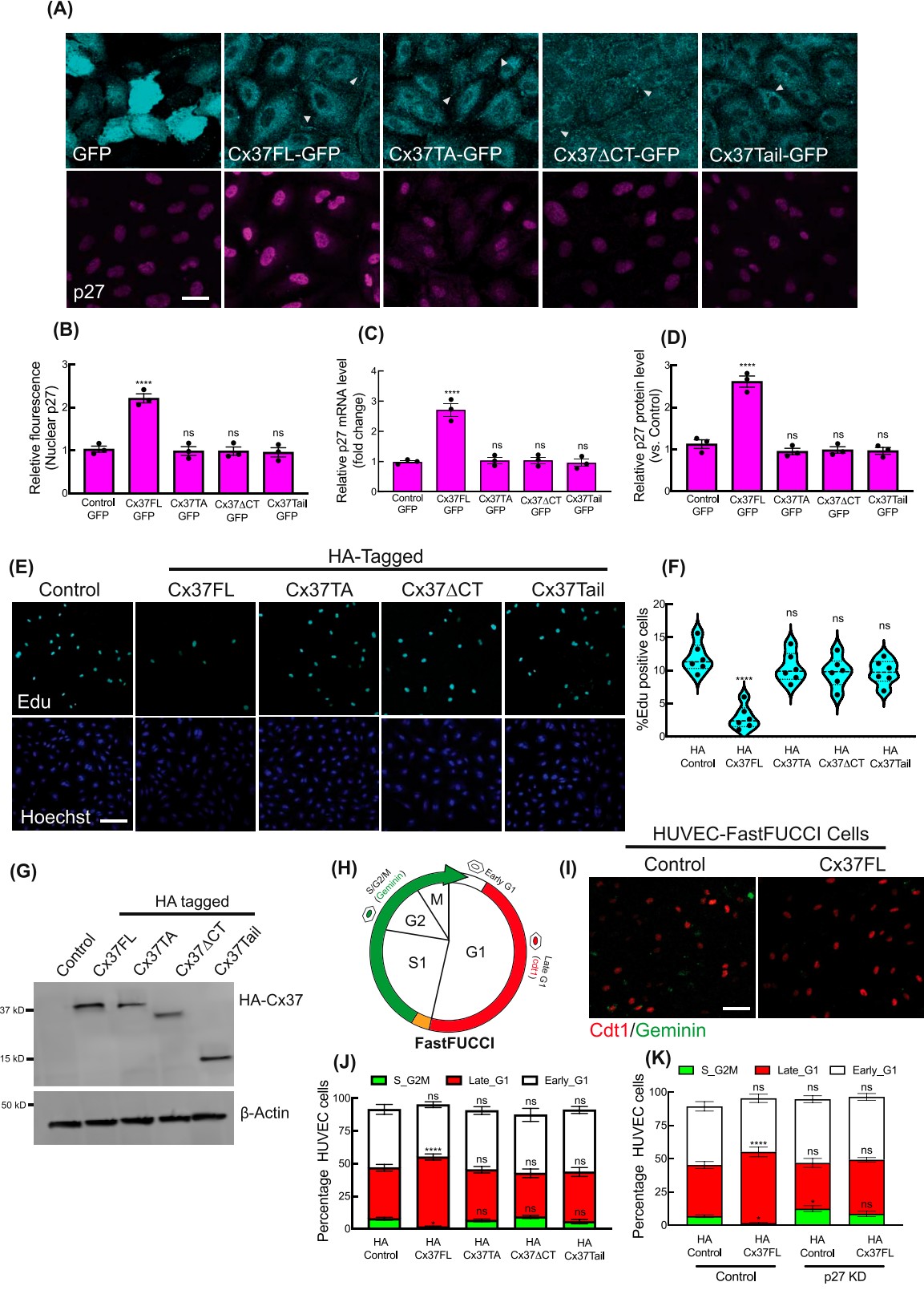

**Figure 1. Regulation of p27 expressions by differential Cx37 constructs.**
**(A)** Localization of GFP-tagged Cx37 constructs and corresponding p27 immunofluorescence in HUVEC. Arrowhead indicating the localization of Cx37 constructs on the membrane and at cell–cell junctions. **(B, C, D)** Quantification of p27 immunofluorescence and the expression profile of p27 mRNA and protein in Cx37-GFP constructs in HUVEC, values normalized to control (GPF-empty vector-expressing HUVEC). **(E, F)** Edu fluorescence and % of EdU-positive cell quantitation (F) indicating the effect of

cell cycle regulatory kinases or transcription of cell cycle modulators have not been studied.

Here, we investigate the mechanism(s) by which Cx37 regulates p27, and thereby endothelial cell cycle control, using HUVEC that express the fluorescent ubiquitination cell cycle indicator (FastFUCCI) (24). Earlier, we found that arterial FSS induces Cx37 expression in HUVEC via Notch activation (4); therefore, we similarly elevated the expression level of Cx37 in HUVEC via exogenous expression of wild type Cx37, in the presence of endogenous Cx37, to delineate its mechanism of p27 regulation and cell cycle arrest. We show that Cx37 expression up-regulates p27 and induces late G1 arrest; conversely, Cx37 knockdown decreases p27 expression and increases the proportion of cells actively cycling in S/G2/M. Furthermore, cytoplasmic sequestration of activated ERK by Cx37 C-terminal domain prevents the efflux of its nuclear target Foxo3a, a transcription activator that up-regulates p27 to promote EC growth arrest (25). Consistent with previous studies, we found this Cx37/pERK/Foxo3a/p27 signaling axis is activated downstream of arterial FSS to promote EC late G1 state and enable arterial gene upregulation.

These results reveal a previously unknown molecular link between Cx37 and pERK signaling, which also governs EC growth suppression and arterial specification. Insights gained can be harnessed for clinical and therapeutic applications.

## Results

### Cx37 promotes endothelial cell late G1 arrest via p27 up-regulation

An applied FSS (18 dyn/cm$^2$ for 12 h) on HUVEC, as predicted (4), augmented Notch-mediated expression of Cx37 (Fig S1A–C); Cx37, in turn, up-regulated p27 expression (Fig S1B and D). To delineate the mechanism(s) by which Cx37 mediates p27 up-regulation, we expressed full-length mouse Cx37-GFP (Cx37FL) in HUVEC, in the presence of endogenous Cx37, which was localized to the plasma membrane and the cytoplasm (Figs 1A and S1E). Cx37FL expression increased p27 mRNA and protein levels (Figs 1B–D and S1F) compared with the empty vector (henceforth termed Control, unless otherwise defined). However, the expression of two Cx37 C-terminal domain mutants: (1) Cx37 without C-terminal tail (Cx37ΔCT; truncated at aa273; localized to PM), and (2) Cx37 C-terminal tail only (Cx37Tail; aa233 to aa333; cytosolic) did not up-regulate p27 expression (Figs 1A–D and S1F). In addition, expression of a Cx37 poreforming domain mutant (22) Cx37T154A (Cx37TA) in HUVEC also failed to up-regulate p27 (Figs 1A–D and S1F). These data indicate that Cx37 C-terminal domain and its channel function are important for p27 up-regulation. Unlike p27, other members of the Kip family of cell cycle inhibitors, p21 and p57, were not up-regulated by Cx37

overexpression (Fig S1G). Consistent with that, when endogenous Cx37 was knocked down with doxyciline (DOX)-inducible shRNA in HUVEC (Fig S1H and I), only the reexpression of Cx37FL, but not any other Cx37 mutants, could restore p27 expression (Fig S1J–L). Of note, the antibodies used for the detection of endogenous Cx37 in GJs in HUVEC also showed a nonspecific nuclear staining, which was unaffected by Cx37 knockdown (Fig S1I). Notably, overexpression of only HA-tagged Cx37FL in HUVEC in endogenous Cx37 background suppressed EdU uptake in HUVEC, indicating inhibited DNA synthesis (Fig 1E and F). Unfrangmented nuclei in the transfected HUVEC indicated that expression of the Cx37 constructs in HUVEC did not induce cell death, which was further confirmed by MTT analysis (Fig S2A). In addition, Lucifer yellow dye transfer studies in HUVEC expressing the Cx37 contructs revealed that HA-Cx37FL and HA-Cx37ΔCT overexpression modestly, but significantly, increased gap junction intracellular communications (GJIC), which did not occur in HUVEC-expressing Cx37TA (Fig S2B and C). Notably, overexpression of any Cx37 construct in HUVEC did not reduce the basal level of HUVEC GJIC (Fig S2C).

We further evaluated Cx37 effects on cell cycle state using HUVEC-FastFUCCI, in which cells are unlabeled in early G1, express Cdt1-mKO2 (red) in late G1 nuclei, exhibit yellow nuclei in G1-to-S transition, and express nuclear Geminin-mAG (green) in S/G2/M (24) (Fig 1H). We and others have shown that FastFUCCi cells in distinct cell cycle states, including early G1 versus late G1, exhibit significant differences in gene and protein expression, and molecular regulation (5). Compared with untransfected control HUVEC, those expressing HA-Cx37FL (Fig 1G), in the endogenous Cx37 background, showed an increased proportion of red nuclei (Fig 1H and I), indicating increased late G1 arrest. Flow cytometric analysis of cell cycle state of HUVEC-FastFUCCI (Figs 1J and S2D) expressing wild type and mutant HA-Cx37 constructs (Fig 1G) revealed that only HA-Cx37FL increased the proportion of HUVEC-FastFUCCI in late G1, with a significantly reduced proportion in S/G2/M (Fig 1J). p27-KD in HUVEC-FastFUCCI abrogated the effects of Cx37 on cell cycle distribution (Figs 1K and S2E and F), indicating Cx37 promotes late G1 arrest via p27, and that Cx37 GJ channel activity and c-terminus are required to promote p27-mediated endothelial growth arrest.

### Cx37 expression retained nuclear Foxo3A that promotes p27 expression and late G1 arrest

Because Cx37 overexpression significantly increased p27 mRNA expression (Fig 1C), we investigated whether Cx37 up-regulates p27 transcription via forkhead box class O family (Foxo) proteins; a known transcription activator of CDKN1B (encodes p27) (25). The effects of Cx37FL overexpression, in the endogenous Cx37 background, on Foxo proteins was examined and we found that Foxo3a protein level was increased in HUVEC, but Foxo1 and Foxo4 levels were unchanged (Fig 2A–C). Notably, Cx37FL expression did not

---

different Cx37 constructs' expression on cell proliferation. **(G)** Expression of HA-tagged Cx37 constructs. Control indicating the HUVEC-expressing HA-empty vector. **(H)** Schematic image of HUVEC-FastFUCCI reporter distinguishes cell cycle stages. **(I)** HA-Cx37FL expression increases cdt1-RFP-positive nucleus population in HUVEC-FastFUCCI cells compared with the HA-control cells. **(J)** HA-Cx37 constructs expression alters HUVEC-FastFUCCI cell cycle states. **(K)** Effect of HA-Cx37FL overexpression on p27-KD HUVEC-FastFUCCI cell cycle. **(B, C, D, J, K)** One-way ANOVA (B, C, D, F) with Dunnett's multiple comparisons test, two-way ANOVA (J, K) with Sidak's multiple comparisons test. Scale bar: 20 $\mu$m (A) and 100 $\mu$m (E, I).

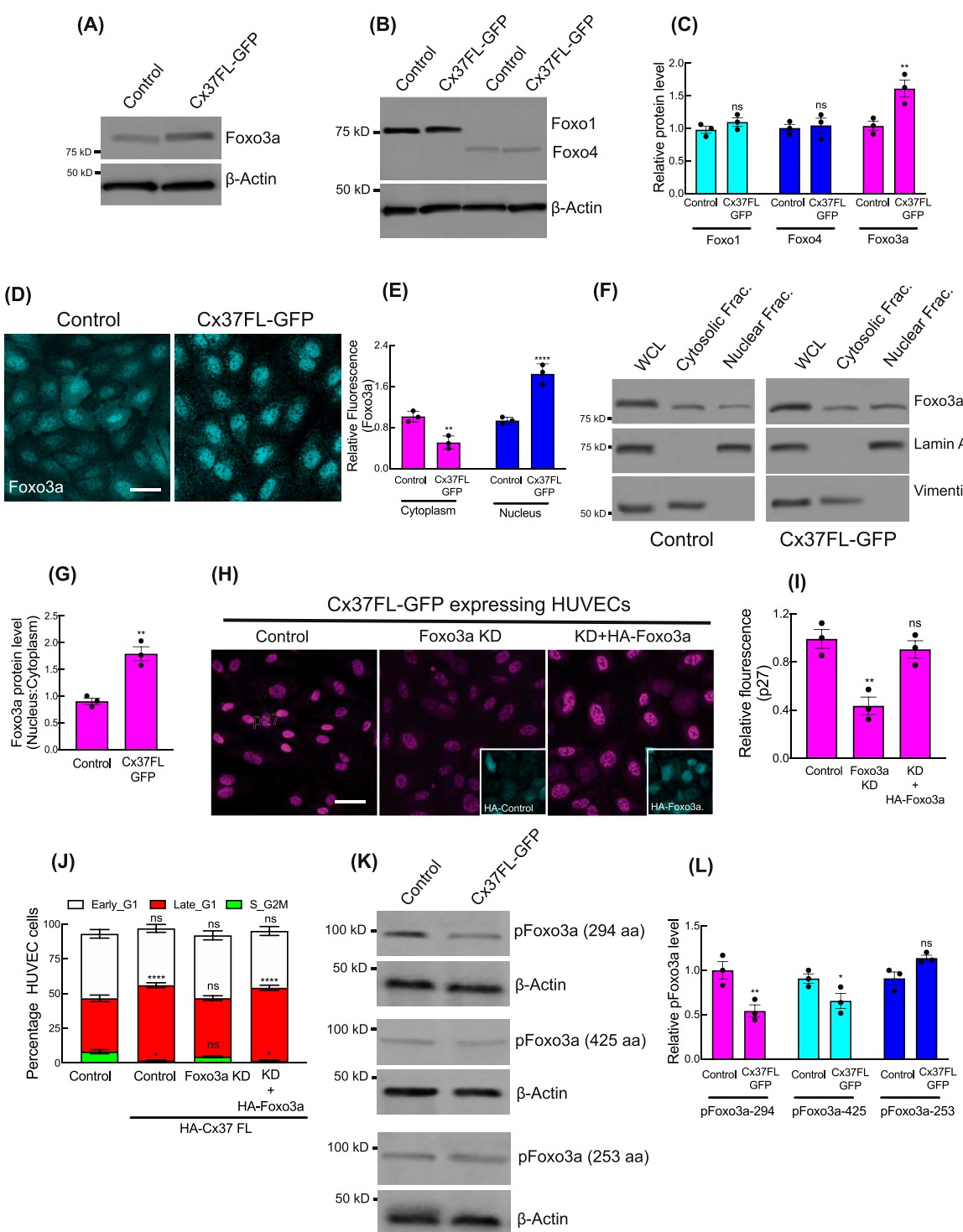

**Figure 2. Nuclear stabilization of Foxo3a by Cx37 and p27 up-regulation.**
**(A, B, C)** Immunoblot showing the protein levels of different Foxo proteins in Cx37FL-expressing HUVEC, and relative quantitation of the protein level compared with GFP-empty vector (control)-expressing cells (C). **(D, E)** Immunofluorescence (D) and relative nuclear fluorescence quantitation (E) of Foxo3a in Cx37FL-expressing HUVEC. **(F, G)** Foxo3a protein level in nuclear versus cytosolic cell extract; immunoblot for Foxo3a, lamin A (nuclear control) and vimentin (Cytosolic control) and quantitation (G). **(H)** Immunofluorescence of p27 in Cx37FL-expressing cells and the effect of Foxo3aKD and rescue with HA-Foxo3a. Foxo3aKD cells are expressing Ha-Empty vector. **(J)** Foxo3a KD affected the cell cycle state of Cx37FL-expressing HUVEC-FastFUCCI. **(K, L)** Phospho-Foxo3a immunoblots showing the effect of Cx37FL-GFP contruct on Foxo3a phosphorylation when control cells are expressing GFP-Empty vector. **(C, E, G, I, J, L)** Unpaired *t* test with Welch correction (G), one-way ANOVA (I) with Dunnett's multiple comparisons test, two-way ANOVA (C, E, J, L) with Sidak's multiple comparisons test. Scale bar: 20 *μ*m (D, H).

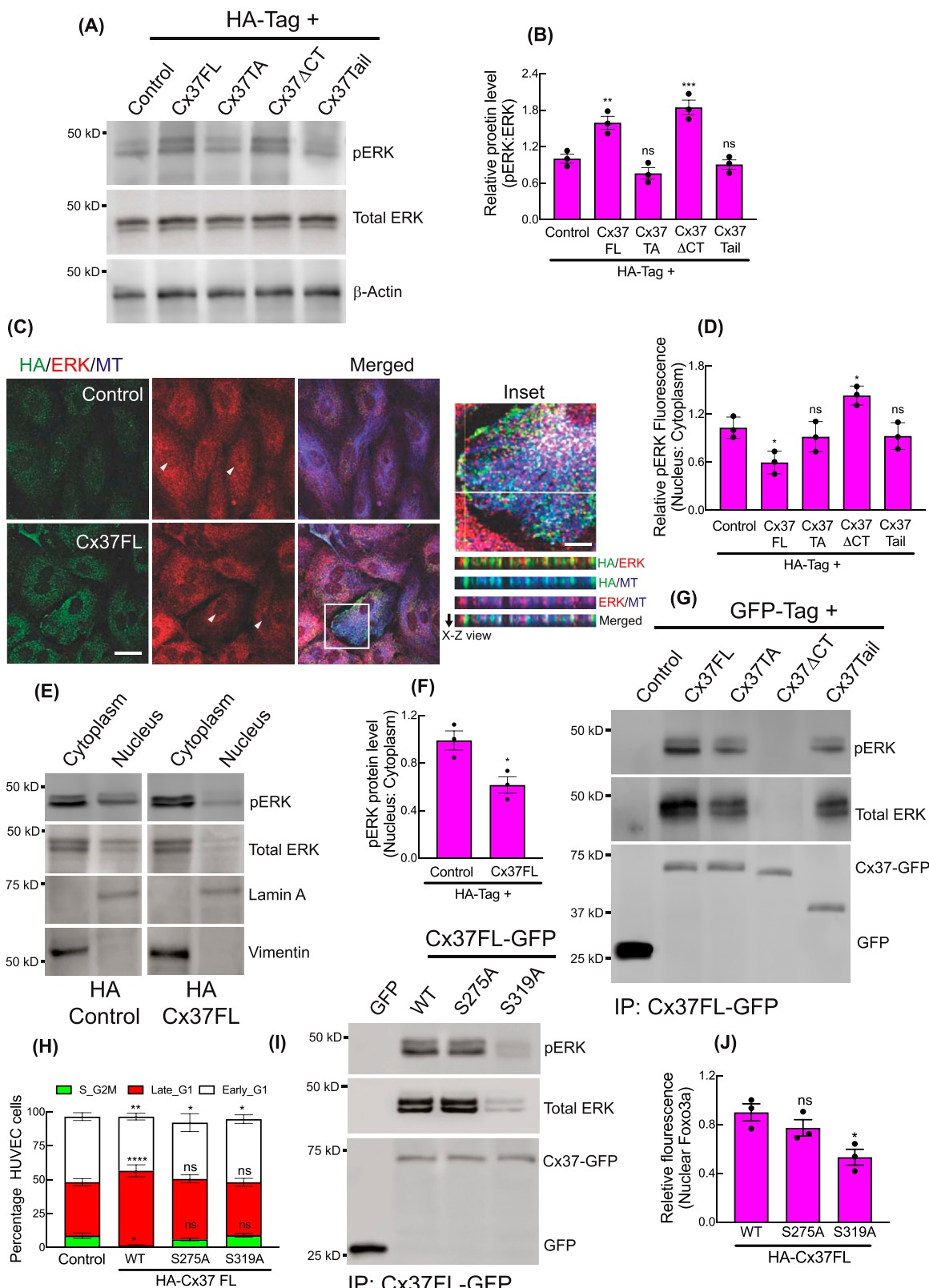

**Figure 3. Cytosolic sequestration of Activated ERK by Cx37.**
**(A, B)** ERK and pERK protein levels in Ha-Cx37 construct expressing HUVEC; immunoblot (A) and quantitation (B), Control denoting the expression of HA-Empty vector. **(C)** Localization of ERK and Cx37 on cytosolic and cortical MT in HA-Cx37FL and HA-Empty vector expressing HUVEC. Inset showing the orthogonal co-localization of Cx37 with ERK and MT along the X–Z axis (white line). Arrowhead indicating the fluorescence of nuclear ERK population in cells. **(D)** Quantitation of relative fluorescence of pERK in HA-Cx37-expressing cells. **(E, F)** Relative

alter mRNA expression for Foxo3a, and another known p27 transcription activator, MEN1 (encodes MENIN) (Fig S3A and B). However, anti-Foxo3a immunofluorescence revealed an increased nuclear localization of Foxo3a in Cx37FL-expressing HUVEC (Fig 2D and E), and immunoblotting further confirmed Foxo3a elevation in the nuclear fraction of these cells compared with the cytoplasm (Fig 2F and G), suggesting that Cx37 stabilizes Foxo3a protein in the nucleus.

Foxo3a-KD (Fig S3C and D) significantly impaired the ability of Cx37FL to up-regulate p27 expression (Figs 2H and I and S3E and F) and promote late G1 arrest (Fig 2J). Reexpression of HA-Foxo3a rescued these effects (Figs 2H–J and S3E and F), indicating that Foxo3a is required for Cx37-induced p27 up-regulation. Concurrently, Cx37FL-induced nuclear localization of Foxo3a was associated with down-regulation of c-MYC (26); a major transcription repressor of p27 (27) (Fig S3G–I), and promoter of arterial gene expression (3). When Foxo3a was inhibited in Cx37FL-expressing HUVEC, c-MYC expression was unaffected (Fig S3H and I).

Activated kinases (phospho-PI3K/AKT/ERK) are known to phosphorylate Foxo3a in the nucleus and promote its efflux to the cytoplasm for degradation (28, 29, 30). Phospho-immunoblotting revealed a significant reduction in ERK-specific phosphorylation sites of Foxo3a (S294, S425) in Cx37FL-expressing HUVEC; in contrast, the AKT-specific site (S253) was unaffected (Fig 2K and L). In addition, the ERK inhibitor U0126, but not PI3K inhibitor LY29400, significantly increased nuclear Foxo3a (Fig S3J and K). These results suggest that ERK signaling may play a role in the regulation of Foxo3a-mediated Cx37 up-regulation of p27 in ECs (31).

### Cx37 C-terminal tail interacts with pERK and impedes its nuclear translocation

Phosphorylated ERK (pERK) translocation to the nucleus is required for its effects on cell proliferation; in the G1 phase, it suppresses cell cycle inhibitors to facilitate G1-to-S phase transition (32). We observed an elevation of pERK in HUVEC-expressing Cx37FL and Cx37ΔCT, in the presence of endogenous Cx37, but not Cx37TA, and total ERK expression was unaffected in all conditions (Fig 3A and B).

Interestingly, immunofluorescence revealed increased accumulation of pERK and ERK in the cytoplasm of Cx37FL-expressing HUVEC in contrast to control (Figs 3C and D and S4A), where they were uniformly distributed in the cytosol and nucleus. Furthermore, the cytosolic fraction of Cx37FL-expressing HUVEC contained more pERK (Fig 3E and F), suggesting cytoplasmic retention of pERK by Cx37. Indeed, we found that Cx37 co-immunoprecipitated pERK in all Cx37-GFP-expressing HUVEC, except those expressing Cx37ΔCT (Fig 3G), suggesting that the Cx37 C-terminal tail interacts with pERK. We next investigated how Cx37 overexpression triggers ERK

phosphorylation in HUVEC. Previous studies suggested that ERK spatial localization, substrate binding, and activation are regulated by intracellular calcium (Ca2+), and it is know that Cx-mediated gap junctional intercellular communication increases intracellular Ca2+ in the endothelium, which modulates vascular tone (33). We therefore investigated whether intracellular Ca2+ is affected by Cx37 overexpression. Using a rhodamine-labelled Ca2+ indicator (AM ester), we observed an increase in cytosolic-free Ca2+ in Cx37-FL-expressing cells (Fig S4B and C), which was strikingly reduced in Cx37-TA-expressing cells, indicating the Cx37 channel function is necessary for Ca2+ influx in HUVEC. Cells treated with a calcium chelator (BAPTA-AM) exhibited reduced pERK in Cx37-expressing cells (Fig S4D and E). These data suggest that increased cytosolic Ca2+ promotes ERK activation/phosphorylation and its interactions with Cx37.

Intriguingly, Cx37 and ERK appeared to be colocalized on microtubules (MT) (Fig 3C inset, X-Z view), and Cx37FL, but not Cx37ΔCT, was able to co-imunoprecipitate α-Tubulin (Fig S4F). This is consistent with previous findings that both connexin and ERK are capable of MT binding (34, 35). Interactions of Cx37 with MT and pERK was further evaluated using the proximity ligation assay (PLA, NaveiFlex in situ proximity ligation technology), which only detects in situ protein–protein interactions when two proteins are within <40 nm (36). MT (Fig S4G and H) and pERK (Fig S5A and B) both showed higher PLA signals with Cx37 in Cx37FL-expressing HUVEC, compared with controls. However, both MT-Cx37 and pERK-Cx37 PLA signals were lower in Cx37ΔCT-expressing cells, suggesting that this interaction is mediated via Cx37 C-terminal tail (Figs S4H and S5B). Similarly, pERK and Cx37 interaction was also impaired in the presence of Nocodazole, which depolymerizes MT (37), suggesting that MT may foster interactions between Cx37 and pERK (Fig S5C and D). Also, ERK interaction with Cx37FL (Fig S5E and F) was also inhibited when cells were treated with MEK-inhibitor U0126, suggesting that only the phosphorylated ERK population interacts with Cx37.

To investigate whether pERK phosphorylates Cx37 as a substrate, we used PhosphoSite.org to determine whether the Cx37 protein sequence contains ERK consensus sites (38), either the SP/TP motif (minimal motif) or PXSP/PXTP motif (full motif). We identified two ERK consensus motifs with putative ERK phosphorylation sites, S275 and S319 (21), on the C-terminal of Cx37 (Fig S5G). We generated phosphodeficient mutants, Cx37FL$_{S275A}$ and Cx37FL$_{S319A}$; however, neither of these mutants could induce late G1 arrest (Fig 3H), and only Cx37-FL$_{S319A}$ was unable to pull-down endogenous pERK (Fig 3I), and thus reduce nuclear Foxo3a and p27 (Figs 3J and S5H). Of note, the in vitro ERK kinase assay with unmodified Cx37 synthetic peptides revealed phosphorylation at only S275, and not at S319 (21), (Fig S5I). Altogether, these results suggest that Cx37 phosphorylation by pERK, and binding of pERK to Cx37 on C-terminal tail, are crucial for cell cycle regulation and growth arrest.

---

abundance of pERk protein in nucleus fraction compared with the cytosolic in HA-Cx37FL-expressing HUVEC; immunoblot (E), and quantitation (F). **(G)** Immunoblot after GFP-trap immunoprecipitation showing the co-immunoprecipitation of ERK and pERK with different Cx37-GFP constructs. **(H)** HA-Cx37FL-phosphmutants alter cell cycle state of HUVEC-FastFUCCI. **(I, J)** Immunoblot after GFP-trap immunoprecipitation and quantitation (J) showing Cx37FL-GFP phosphmutants alter ERK and pERK co-immunoprecipitation with Cx37. **(B, D, F, H, J)** Unpaired *t* test with Welch correction (F), one-way ANOVA (B, D, J) with Dunnett's multiple comparisons test, two-way ANOVA (H) with Sidak's multiple comparisons test. Scale bar: 20 *μ*m ((C); and 5 *μ*m inset).

## Cytosolic ERK sequestration and nuclear Foxo3a mediate FSS-promoted arterial gene expression

We further investigated whether the Cx37/pERK/Foxo3a/p27 signaling axis was activated and required for FSS-induced G1 arrest and arterial gene up-regulation (4, 39). Immunofluorescence and pERK/ERK immunoblotting revealed that arterial FSS increased both nuclear Foxo3a and cytosolic pERK in HUVEC, in comparison with control cells in static culture conditions without FSS stimulation. These effects were suppressed when Cx37 was inhibited (Figs 4A–F and S6A).

Arterial FSS-induced late G1 arrest in HUVEC-FastFUCCI was abrogated by Foxo3a knockdown and ERK/MEK inhibition (Fig 4G). Expression of Cx37FL, but not Cx37ΔCT, in HUVEC-FastFUCCI, in which Cx37 was knocked down, restored FSS-induced late G1 arrest (Fig S6B) and up-regulated arterial genes *Hey2*, *Efnb2*, and *GJA5*, but not venous genes *Ephb4* and *Nr2f2* (Figs 4H and S6C). Similarly, Foxo3a inhibition reduced FSS-induced arterial gene up-regulation (Fig 4I), except *Hey2*, which is activated directly by FSS-induced Notch signaling. Similar to HUVEC, in human aortic endothelial cells (HAEC) expressing FastFUCCI (HAEC-FastFUCCI), we observed a similar induction of late G1 arrest in response to arterial FSS (Fig S6D and E), which was abrogated when Notch signaling, and ERK activation, were inhibited by DAPT and U0126, respectively (Fig S6E). Also, both Cx37 and Foxo3a knockdown inhibited FSS-induced late G1 arrest in HAEC-FastFUCCI (Fig S6E), as we observed in HUVEC-FastFUCCI.

In HAEC, with endogenous Cx37 knocked down, arterial gene expression was up-regulated by arterial FSS in the presence of Cx37FL, but not Cx37ΔCT (Fig S6F). Similar to HUVEC, arterial FSS did not affect venous gene expression in HAEC. These data suggest that the Cx37/pERK/Foxo3a signaling axis functions downstream of arterial FSS to mediate late G1 arrest of endothelial cells and promote arterial gene expression in vascular endothelial cells.

# Discussion

Our results delineate a mechanism by which Cx37 mediates p27 up-regulation and its ability to promote late G1 arrest in ECs, which is required to enable arterial gene expression. We found that Cx37 sequesters pERK in the cytoplasm, which enhances Foxo3a nuclear retention and promotes p27 transcription (25). Furthermore, we found that this Cx37/pERK/Foxo3a/p27 signaling axis functions downstream of arterial shear stress.

More specifically, Cx37 overexpression and arterial shear stress enhanced nuclear localization of Foxa3 to enable p27 transcriptional up-regulation (25) and inhibition of c-MYC (26). We found this Foxo3a-mediated p27 up-regulation is ERK dependent, as we showed that the Cx37 C-terminus binds to pERK and sequesters it in the cytoplasm to prevent its inactivation of Foxo3a. There also appears to be reciprocal regulation of Cx37 by pERK; that is, inhibition of ERK phosphorylation on the Cx37 C-terminal tail prevents the ability to promote late G1 arrest.

The C-terminal domain of connexin 43 (Cx43) was previously shown to regulate Akt/ERK hyperphosphorylation through direct interactions (40). However, our studies suggest a different mechanism for Cx37; that is, the Cx37 C-terminal domain interacts in a complex with pERK and MT, which is necessary for pERK sequestration in the cytoplasm. Our data further suggest that Cx37 up-regulation contributes to ERK phosphorylation by increasing intracellular-free $Ca^{2+}$, which is an upstream regulator of ERK activation (41, 42, 43) and it can be transferred through GJs and hemichannels composed of almost all connexins (42).

Similar to Cx37, induced expression of other connexins have been shown to propagate intercellular $Ca^{2+}$ waves, but whether that is because of $Ca^{2+}$ influx from outside the cell, or release from the ER membrane, is not clear. An intracellular $Ca^{2+}$ abundance after Cx43 overexpression was shown to be abrogated by inhibition of GJ channel function, but not by inhibiton of $Ca^{2+}$ release from ER membranes, suggesting a functional role for the Cx43 GJ channel (44, 45). Our observations are consistent with this, and show that Cx37FL expression, but not Cx37TA, increases intracellular-free $Ca^{2+}$, which supports an active role of channel-forming domain in this signaling pathway. Perhaps the Cx37 channel domain functions to mediate the passage of $Ca^{2+}$ through GJs and hemichannels to elevate intracellular $Ca^{2+}$ or to liberate $Ca^{2+}$ from a spatially sequestered puff to promote ERK activation (46). Nevertheless, the interplay between connexin channel function and $Ca^{2+}$ signaling is complex and needs further intensive investigation (42, 43, 47).

ERK activation and hemodynamic stress both promote arterial specification in remodeling vessels (48, 49). Here, we found these mechanisms are connected; that is, arterial shear stress increased cytosolic pERK downstream of Cx37, and ERK inhibition suppressed shear stress-induced late G1 arrest. In addition, deletion of the C-terminal tail of Cx37 that binds pERK prevented its up-regulation of arterial genes in response to shear stress. Thus, our studies suggest that this Cx37/pERK/Foxo3a/p27 axis is activated in response to arterial shear stress to promote arterial gene expression.

# Materials and Methods

## Cell lines and cell culture details

### Primary HUVEC were obtained from Yale

Core facility was only used up to passage 8 for experiments. Cells were passaged in Endothelial Cell Growth Medium EGM-2 (Cat# CC-

**Key Resource Table.**

| Reagents or resource | Source | Identifier |
|---|---|---|
| Recombinant DNAs | | |
| pLJM1-EGFP | Addgene (Sabatini Lab) | Cat#19313 |
| pULTRA | Addgene (Moore Lab) | Cat#24129 |
| Tet-pLKO-Puro | Addgene (Weiderschain lab) | Cat#21915 |

**Continued**

| Reagents or resource | Source | Identifier |
|---|---|---|
| pcDNA3.1-HA | Addgene (Laur lab) | Cat#128034 |
| *H. sapience* HA-Foxo3a WT | Addgene (Greenberg lab) | Cat#1787 |
| pLJM1-mCx37FL-EGFP (m = *Mus musculus*) | This study | N/A |
| pLJM1-mCx37TA-EGFP | This study | N/A |
| pLJM1-mCx37ΔCT-EGFP | This study | N/A |
| pLJM1-mCx37Tail-EGFP | This study | N/A |
| pLJM1-mCx37S275A-EGFP | This study | N/A |
| pLJM1-mCx37S319A-EGFP | This study | N/A |
| pLJM1-mCx37S321A-EGFP | This study | N/A |
| pULTRA-HA-mCx37FL | This study | N/A |
| pULTRA-HA-mCx37TA | This study | N/A |
| pULTRA-HA-mCx37ΔCT | This study | N/A |
| pULTRA-HA-mCx37Tail | This study | N/A |
| pULTRA-HA-mCx37S275A | This study | N/A |
| pULTRA-HA-mCx37S319A | This study | N/A |
| pULTRA-HA-mCx37S321A | This study | N/A |
| Antibodies | | |
| Cx37/GJA4 | Abcam | Cat# ab181701 |
| CX37/GJA4 | Thermo Fisher Scientific | Cat# 40-4300 |
| p27 KIP 1 | Abcam | Cat# ab193379 |
| p27 KIP 1 | Cell Signaling Technology | Cat# 3686 |
| p21 (CIP1/WAF1) | Abcam | Cat# ab218311 |
| p57 KIP2 | Cell Signaling Technology | Cat# 2557 |
| c-MYC | Novus Biologicals | Cat# NB100-1642 |
| Foxo3a | Cell Signaling Technology | Cat# 2497 |
| Phospho-FoxO3a (Ser253) | Cell Signaling Technology | Cat# 9466 |
| Phospho-FoxO3a (Ser294) | Cell Signaling Technology | Cat# 5538 |
| Phospho-FoxO3a (Ser425) | Cell Signaling Technology | Cat# 64616 |
| FoxO1 | Cell Signaling Technology | Cat# 2880 |
| FoxO4 | Cell Signaling Technology | Cat# 9472 |
| P44/42 MAPK (Erk1/2) | Cell Signaling Technology | Cat# 9102 |
| Phospho p44/42 MAPK (Erk1/2) | Cell Signaling Technology | Cat# 9106 |
| GFP-tag | Origene | Cat# R1091TR |
| HA-tag | R&D Systems | Cat# MAB060 |
| VE-Cadherin | R&D Systems | Cat# AF938 |
| Alpha Tubulin (YOL1/34) | Novus Biologicals | Cat# NB100-1639 |
| Beta-Actin | Cell Signaling Technology | Cat# 4967 |
| DNA cloning primers | | |
| Construct | Forward (5′ to 3′) | Reverse (5′ to 3′) |
| pLJM1-mCx37TA-EGFP | CGTCAGATCCGCTAGCGCCACCATGGGCGACTGG | CATGGTGGCGACCGGTCACATACTGCTTCTTGGATGC |
| pLJM1-mCx37ΔCT-EGFP | CGTCAGATCCGCTAGCGCCACCATGGGCGACTGG | CATGGTGGCGACCGGTGGGTCCCTCGCCCATGGG |
| pLJM1-mCx37Tail-EGFP | CGTCAGATCCGCTAGCGCCACCATGGTCAGCCGG | CATGGTGGCGACCGGTCACATACTGCTTCTTGGATGC |

| Reagents or resource | Source | Identifier |
|---|---|---|
| Annealed HA-oligo in pULTRA vector | CCGGTGCCACCATGGTGTACCCATACGATGTTCCAGATTACGCTT | CTAGAAGCGTAATCTGGAACATCGTATGGGTACACCATGGTGGCA |
| pULTRA-HA-mCx37FL | TCCCGGGCCTTCTAGAATGGGCGACTGGGGCTTC | CGCCGGAGCCGGATCCCTACACATACTGCTTCTTGGATGC |
| pULTRA-HA-mCx37ΔCT | TCCCGGGCCTTCTAGAATGGGCGACTGGGGCTTC | TCCCGGGCCTTCTAGAATGGGCGACTGGGGCTTCC |
| pULTRA-HA-mCx37Tail | AGATTACGCTTCTAGATGTCGGTGTGTCAGCCGG | CGCCGGAGCCGGATCCCTACACATACTGCTTCTTGGATGC |
| GJA4-T154A | GCTCATGGGTGCCTATGTGGTCA | GCCCCACGAATCCGAAGA |
| GJA4-S275A | CGAGGGACCCGCTTCCCCACCGT | CCCATGGGGAGGTAGAAGAAAACCTG |
| GJA4-S319A | TGGCCGAAAGGCCCCTAGCCGCC | CCCTGGGGAGCTGTGTTTACAAATGGGG |
| GJA4-S321A | AAAGTCCCCTGCCCGCCCCAACAGCTCTG | CGGCCACCCTGGGGAGCT |
| Human GjA4 shRNA1 | CCGGGAAGCAGAAGATAACACCTCTCTCGAGAGAGGTGTTATCTTCTGCTTCTTTTTG | AATTCAAAAAGAAGCAGAAGATAACACCTCTCTCGAGAGAGGTGTTATCTTCTGCTTC |
| Human GjA4 shRNA2 | CCGGGGGGTGACGAGCAATCAGATTTCTCGAGAAATCTGATTGCTCGTCACCCTTTTG | AATTCAAAAAGAAGCAGAAGATAACACCTCTCTCGAGAGAGGTGTTATCTTCTGCTTC |
| Human Foxo3a shRNA1 (CDS) | Sigma-Aldrich; Mission shRNA | TRC Clone ID: Version 1 TRCN000010335 |
| Human Foxo3a shRNA2 (3'UTR) | Sigma-Aldrich; Mission shRNA | TRC Clone ID: Version 2 TRCN0000235491 |
| siRNA-CDKN1B | Dharmacon Ltd. | Cat# L- 040178-00 |
| siRNA-control | Qiagen | Cat# 1022076 |
| qRT-PCR primers | | |
| Human genes | Forward (5' to 3') | Reverse (5' to 3') |
| *ACTB (β-Actin)* | TCACCCACACTGTGCCCATCTACGA | CAGCGGAACCGCTCATTGCCAATGG |
| *CDKN1B (p27)* | ATCACAAACCCCTAGAGGGGCA | GGGTCTGTAGTAGAACTCGGG |
| *GJA4 (Cx37)* | ACACCCACCCTGGTCTACC | CACTGGCGACATAGGTGCC |
| *FOXO3A* | TTCAAGGATAAGGGCGACAGCAAC | CTGCCAGGCCACTTGGAGAG |
| *c-MYC* | CTTCTCTCCGTCCTCGGATTCT | GAAGGTGATCCAGACTCTGACCTT |
| *MEN1 (MENIN)* | CAGGGGCCAGACAGTCAATG | GGTGGGCTCCAGCTCCTCTA |
| *GJA5 (Cx40)* | CCGTGGTAGGCAAGGTCTG | ATCACACCGGAAATCAGCCTG |
| *HEY2* | GCCCGCCCTTGTCAGTATC | CCAGGGTCGGTAAGGTTTATTG |
| *EFNB2* | TATGCAGAACTGCGATTTCCAA | TGGGTATAGTACCAGTCCTTGTC |
| *NR2F2* | GGACCACATACGGATCTTCCAA | ACATCAGACAGACCACAGGCAT |
| *EPHB4* | CGCACCTACGAAGTGTGTGA | GTCCGCATCGCTCTCATAGTA |
| Reagents and kits | | |
| NaveniFlex MR (PLA Kit) | Cambridge Bioscience | Cat# NF.MR.100 (Suppl. Navinci Diagnostics AB) |
| Dextran, rhodamine B, 70,000 MW, neutral | Invitrogen | Cat# D1841 |
| Lucifer yellow CH, lithium Salt | Invitrogen | Cat# L453 |
| MTT (3-(4,5-Dimethylthiazol-2-yl)-2,5-Diphenyltetrazolium bromide) | Invitrogen | Cat# M6494 |
| Edu Proliferation Assay Kit (iFlour 647) | Abcam | Cat# ab222421 |
| X-rhod-1, AM Cell permeable Ca2+ assay | Thermo Fisher Scientific/molecular probe | Cat# X14210 |

3162; Lonza). The cryopreserved ampule of HAEC was purchased from Lonza (Cat# CC-2535). Cells were passaged in Endothelial Cell Growth Medium EGM-2 (Cat# CC-3162; Lonza) and used only up to passage 6 for experiments. For both HUVEC-FASTFUCCI and HAEC-FUCCI, cells were infected with Fast-FUCCI plasmid (pBOB-EF1-FastFUCCI-Puro; was a gift from Kevin Brindle & Duncan Jodrell; Addgene plasmid # 86849) containing lentivirus generated within HEK293T cells. All Cx37 constructs expressing cells were selected and passaged in puromycin (0.5 $\mu$g/ml, P9620; Sigma-Aldrich) for stable expression of genes. Cultured cells were tested for mycoplasma contamination routinely, and over the course of these studies, no positive test was obtained.

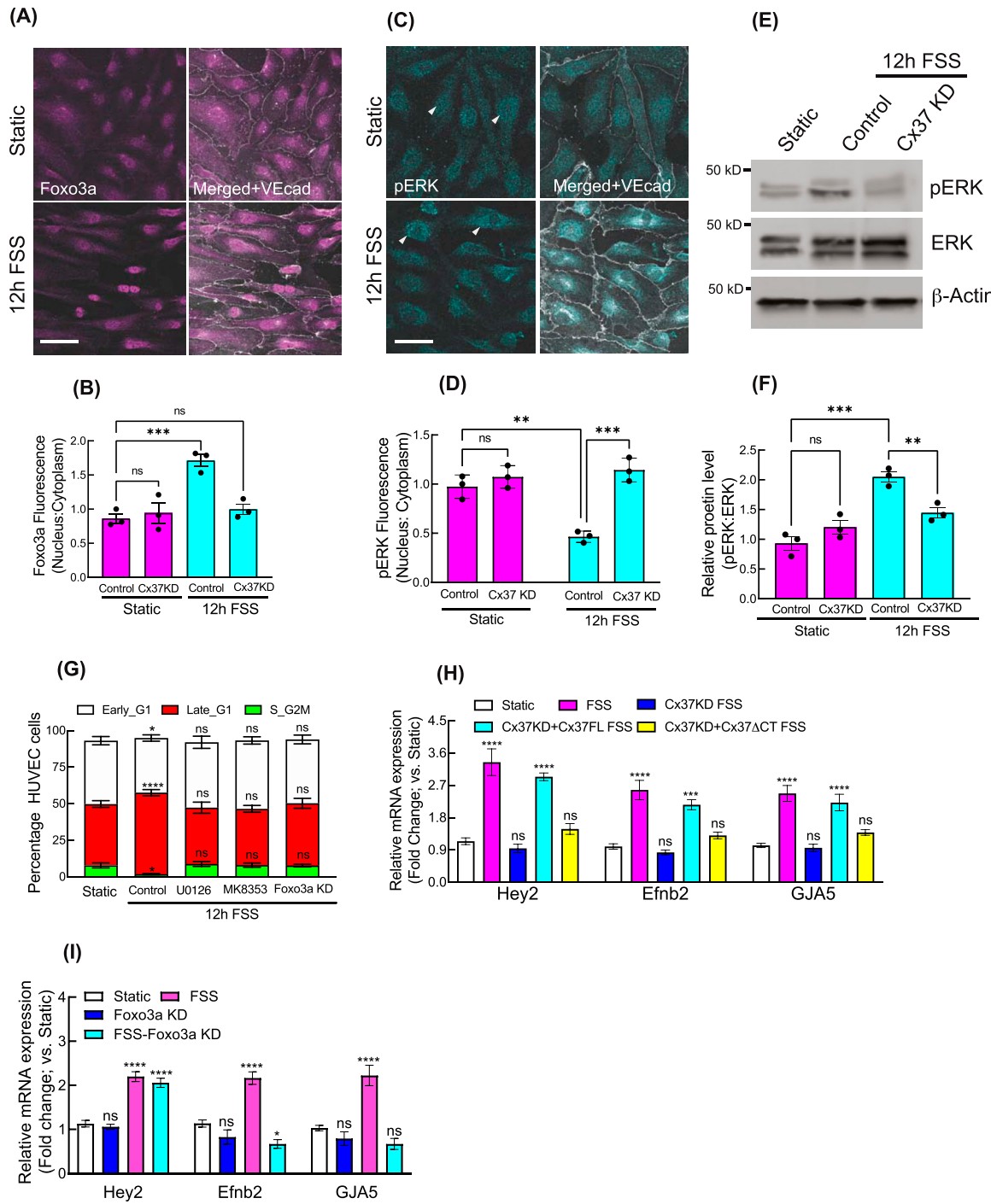

**Figure 4. Cx37/ERK/Foxo3a is operational in FSS-induced cell cycle arrest and arterial fate specification.**
**(A, B, C, D)** Effect of FSS on relative localization of Foxo3a and pERK; immunofluorescence (A, C) and quantification (B, D). **(E, F)** Effect of Cx37 knockdown on ERK and pERK protein level; immunoblot (E) and quantitation (F). **(G)** Effect of Foxo3a KD and kinase inhibitors on FSS-induced HUVEC-FastFUCCI cell cycle. **(H)** In Cx37 KD background, the effect of HA-CX37FL and ERK-binding mutant of Cx37; HA-Cx37ΔCT on arterial genes expression in FSS-induced HUVEC. **(I)** The effect of Foxo3a KD on arterial genes expression in FSS-induced HUVEC. Two-way ANOVA (B, D, F, G, H, I) with Sidak's multiple comparisons test. Scale bar: 40 µm (A, C).

## Cell transfection and lentivirus infection

Target proteins were either depleted by siRNA, cells in EGM-2 media (without antibiotic) were transfected at 80% confluency with 20 nM si-*CDKN1B* or control siRNA for 48 h. For shRNA knockdown of Foxo3a, cells were cultured in EMG-2 media without antibiotics and infected with shRNA containing lentivirus with 10 µg/ml polybrene (Cat# H9268; Sigma-Aldrich). Inducible shRNA knockdown of Cx37 was trigged with or without doxycycline (1 µg/ml, Cat# D9891; Sigma-Aldrich) in the culture medium for 24 h after lentivirus infections.

Cells were incubated up to 48 h in doxycycline for optimal knockdown. Doxycycline inducible-shRNA2 decreased endogenous Cx37 expression in HUVEC more robustly than shRNA1; it was used for immunocytochemical validation of Cx37KD (Fig S1D) and further assays. For Cx37 constructs' expression, 80–90% confluent cells were incubated with the desired lentivirus (grown in HEK293T cells) with 10 μg/ml polybrene; 24–48 h post infection, the cells were processed for experiments and analysis as required. For HA-Foxo3a, the plasmid HA-FOXO3a WT was a gift from Michael Greenberg (Addgene# 1787) and cells was transfected with this plasmid or with control-HA plasmid (pcDNA3.1-HA Addgene# 128034) using Lipofectamine 3000 (Cat# L3000008; Thermo Fisher Scientific), as per the manufacturer's instructions.

### Plasmids

Cx37 constructs, full-length and other domain mutants, were PCR amplified from pTRE2hyg-mouse-Cx37 plasmid (a gift from Jenis Burt; [50]) and cloned in pULTRA-lentivirus vector (a gift from Malcolm Moore; Addgene# 24129) at XbaI/BamHI sites infusion HD-cloning (Takara Biosciences). The pULTRA vector was modified by excision of EGFP and P2A, and replace with the HA tag at AgeI/XbaI sites. For GFP-tagged Cx37 construct, the respective length of DNA was amplified and from pTRE2hyg-mouse-Cx37 and inserted at pLJM1-EGFP vector (a gift from David Sabatini; Addgene# 19319) at NheI/AgeI sites by infusion HD-cloning (Takara Biosciences). All site-directed mutations were generated using Q5-Site-Directed Mutagenesis Kit (NEB), following the manufacturer's protocol. The primers were designed using NEB-Base changer software and the sequences are mentioned in the reagent table. The cloned plasmids were sequenced and confirmed by Sanger sequencing from Eurofin Sequencing Services.

### Reagent and antibodies use

Primary antibodies used in this study are listed the reagent table. Most of the primary antibodies were used for immunofluorescence at the dilution of 1:50 to 1:100 except VEcad (1:400) and α-Tub (1:500). For Western blotting, the primary antibody dilution was 1:500 for all, except Cx37 (1:250). For immunofluorescence, the species-specific secondary antibodies used in this study were conjugated with Alexa Fluor 405, 488, 546, 594, and 647 (1:500; Thermo Fisher Scientific) or with horseradish peroxidase-conjugated secondary antibodies (1:2,500/5,000; Cell Signaling Technologies) for Western blotting.

Cells were treated with DAPT (5 μM for 12 h, Cat#D5942; Sigma-Aldrich), LY294002 (10 μM for 1 h, or 2.5 μM for 12 h; Cat# 1130; Tocris), U0126 (10 μM for 1 h, or 2.5 μM for 12 h; Cat# 9903; Cell Signaling Technology), MK8353 (2.5 μM for 12 h; Cat# SCH900353; SelleckChem), nocodazole (10 μM for 30 min; Cat#M1404; Sigma-Aldrich), and BAPTA-AM (10 μM for 2 h, Cat#A1076; Sigma-Aldrich).

### Fluid shear stress

HUVEC or HEAC cells were seeded into ibidi μ-Slide IV (ibidi GmbH) (1.2 × $10^5$ cells/slides) and incubated at 37°C for 24 h in a 5% $CO_2$ environment. The cells were then subjected to laminar shear stress (18 dyn/cm$^2$) for 12–18 h (as suggested in experiments) using a bidirectional media flow operated by ibidi Pump System (ibidi GmbH). Cellular morphology was observed using light microscopy and fixed with 4% PFA for immunofluorescence study. The relative gene and protein expression levels were measured by harvesting the cells from μ-Slide by trypsinization. Flow cytometry was used for tracking FUCCI reporter expression and to determinate specific cell cycle stages.

### Immunofluorescence, immunoprecipitation, and immunoblotting

For immunofluorescence, cells were either fixed with 4% paraformaldehyde (without methanol) in 1x PBS with 1 mM Ca2+ and 0.5 mM Mg2+ (PBS-CM), and subsequently permeabilized with 0.2% Triton-X in PBS-CM. The cells were blocked and incubated with primary and secondary antibodies in PBS-CM with 5% BSA and 0.1% Tween20. Leica SP8 confocal microscopes (20X or 63X, 1.4NA Plan Apo objective) driven by LAS X Life Science software (Leica) were used for fixed imaging. All image reconstruction and channel alignment were performed within the LAS X software.

For immunoblotting experiments, the total protein of the cell lysates was quantified by BCA assay for Western blots and IP experiments. All immunoblots include a loading control to ensure the uniformity of protein lysate loading for each assay. Whole cell lysates use beta-actin as a loading control, but nuclear versus cytosolic fraction experiments use Lamin A and Vimentin, respectively. For Cx37 GFP-trap immunoprecipitation experiments, HUVEC were cultured on 15 cm culture dishes, at 80–90%% confluence, the cells were infected with lentiviruses with desired Cx37-GFP constructs; after 48 h, the infected cells were harvested and lysed. The clarified cell lysate was incubated with ChromoTek GFP-trap agarose beads. After incubation, GFP-trap beads were washed several times in a lysis buffer supplemented with 300 mM NaCl as directed in the manufacturer's protocol. The protein concentration of every cell lysate was measured and normalized with the control (GFP-empty vector expressing cells) before loading with GFP-trapping agarose beads, the equal volume of final elutes were loaded for immunoblotting. The protein complexes were boiled in the SDS–PAGE loading buffer and then centrifuged at 16,000$g$; 10 min, and resolved by 4–15% Criterion Tris–HCl SDS–PAGE (Cat# 3450028; Bio-Rad) and immunoblotted. Western blots were imaged with the Azure Biosystems c300, and quantified by ImageJ densitometry analysis.

### Image processing and analysis

Quantitative analysis of cytoplasmic, junctional, and nuclear intensities of Cx37 (both endogenous and overexpress constructs), p27, Foxo3a, ERK, pERK, and others were performed in ImageJ software (NIH). Mean cytoplasmic and nuclear fluorescence intensity was measured by masking the entire cytoplasm and nucleus of cells individually and calculating the average pixel intensities within that mask. Often, optical Z-stacks (0.3 mm intervals) were acquired to correct for cell heights and to focus on all cells analyzed. The average pixel intensities of the masks on different region of the cells were recorded and plotted, the intensity profiles were fitted to a Gaussian curve and the peak values (peak fluorescence intensities) were obtained from this fitting. The average pixel intensity values of the nonfluorescent section of the images were considered as background

fluorescence and uniformly subtracted from the intensity plots of images of different conditions. Data are presented as the ratio values normalized to the corresponding ratio value for control conditions. The normalized ratio of nucleus versus cytoplasmic fluorescence intensity is referred to as nuclear p27, Foxo3a or the vice-versa as the cytoplasmic intensity of different proteins.

### Quantitative PCR and EdU proliferation assay

Qiagen RNeasy Mini Kit was use to purify RNA from HUVEC, both at static or under flow, was then converted to cDNA using the High-Capacity cDNA Reverse Transcription Kit (Cat# 4368814; Thermo Fisher Scientific). The cDNA was quantified by Power SYBR Green PCR Master Mix (Cat# 4368577; Thermo Fisher Scientific) using qRT–PCR platform (QuantStudio 6; Applied Biosystems). Gene-specific primers are listed in the reagent table. Relative gene expression was quantified and determined by the Delta Delta Ct method.

For EdU proliferation assay, Abcam EdU Staining Proliferation Kit (iFlour 647; Cat# ab222421) was used, and the manufacturer's protocol was followed. Cells expressing different Cx37 constructs were incubated with EdU solutions under optimal growth condition, fixed cells were incubated with iFlour 647 to fluorescently label-incorporated EdU, and then analyzed by fluorescence microscopy.

### Assessment of gap junctional intercellular communication

Scrape loading dye transfer assay was performed using Lucifer yellow dye to determine the effect of Cx37 overexpression in HUVECs (51). Cells were rinsed with PBS-Cm and loaded with a solution containing Lucifer yellow (1 mg/ml), rhodamine Dextarn (1 mg/ml) and Hoechst 33342 (0.05 mg/ml; Sigma-Aldrich). After the dye is added to the cells; two parallel cuts on randomly selected central part of the dishes were made using curve edge microsurgery blades. After a 10-min incubation, to allow the dye to travel through several adjacent cell layers, the cells were washed with PBS-CM to remove background fluorescence and fixed in 4% paraformaldehyde. Using 20x objective, we acquired simultaneous images for LY/Rh-Dextran/Hoechst in 488 channel, 590 channel, and 405 channel, respectively. For calculating the net LY distance transfer from scratch, the distances of LY and RhD fluorescence from scratch are subtracted from each other for each field of view (i.e., Distance$^{LY}$-Distance$^{RhD}$). Net LY transfer distance of every experimental image can then be normalized to the averaged net LY transfer from control dishes to obtain the fraction of the control (GJIC-FOC) (52).

$$GJIC\ FOC_{Expt.} = \left(Dist.^{LY}_{\ Expt.} - Dist.^{RhD}_{\ Expt.}\right) / \left(Dist.^{LY}_{\ Contl.(Avg.)} - Dist.^{RhD}_{\ Contl.(Avg.)}\right)$$

### MTT assay for determination of cell viability

The cytotoxicity of Cx37 constructs overexpressed in HUVEC was analyzed by MTT assay. HUVECs were incubated for 2 h at 37°C into MTT solution (5 mg/ml in PBS-CM; cell medium and MTT solution proportion was 10:1). Subsequently, 100 $\mu$l of DMSO (dimethyl sulfoxide) into each one well to dissolve the formazone mixture

(30 min at 37°C), and the well plate was incubated with 5% $CO_2$ for 1 h at 37°C. The OD was measured in a plate reader at 560 nm.

### Flow cytometry of FUCCI cells

Experimental HUVEC and HAEC-FUCCI cells were lifted by trypsinization. Subconfluently (70–80%) grown cells were then washed and resuspended in FACS buffer or PBS with 0.1% BSA. Fluorescent levels of mCherry and mVenus were used to determine the cell cycle state during flow cytometry analysis at BD FACSAria or BD FACSMelody.

### NivaniFlex mouse–rabbit Proximity Ligation Assay

Cultured HUVEC on glass coverslips was fixed with 4% PFA and permeabilized with 0.2% Triton-X/PBS-CM. The cells were blocked with 1x NivaniFlex blocking buffer and incubated with diluted (in 1x primary antibody dilutant) rabbit–mouse primary antibody pair; specific to the protein pair as indicated in the figures. After overnight incubation, the coverslips were washed and re-incubated with secondary rabbit–mouse IgG with probes diluted in 1x probe diluent and incubated for 1 h at 37°C. After that, reactions A, B, and C were performed as instructed in the manufacturer's protocol; we used Buffer C (1x; Atto 488) for detecting the interaction-PLA dots in our experiment. We co-stained our coverslips with DAPI and imaged them using 40x objective on a Zeiss Airyscan Confocal 880 system. The Brightfield images were captured for each field to detect the individual cell boundary. PLA dots was quantitated per cells per conditions, and at least 150 cells were counted for every condition.

### In vitro kinase assay and LC-MS/MS analysis

Peptide corresponds to either Cx37 tail region or ERK-consensus site; FYLPMGEGPSSPPCP (Cx37 265–279, mCx37-Peptide1), GRKSPSRPNSSASKK (Cx37 316–330, mCx37-Peptide2) and PRSARPLSPQNSPTG (ERK control peptide) were synthesized by GenScript USA. For in vitro kinase assay, 0.5 mM peptide substrates were incubated with 0.05 $\mu$g/$\mu$l recombinant ERK2 (Cat# 1230-kS-010; R&D systems) and 0.5 mM ATP (Cat# 9804; Cell signaling Technology) in 1X Kinase Buffer (Cat# 9802; Cell signaling Technology) at 30°C for 45 min. The reaction products were analyzed by LC–Mass Spectrometry.

Each peptide was injected separately for data-dependent (DDA) mass analysis on an Ultimate3000 nanoLC integrated to a Thermo Fisher Scientific Orbitrap Eclipse Tribrid system (2.5 pmol injection, PepMap100 C18 column, 75 $\mu$m × 25 cm with 1.5 h RP-HPLC gradient and 300 nl/min flow rate) before kinase assay. Expected peptide masses were confirmed from high-resolution Full MS scans. Outputs from the kinase assay were desalted using Pierce C18 Spin Columns (89870) from Thermo Fisher Scientific per the manufacturer's protocol. The final eluent was dried via speed vac and reconstituted in 50 $\mu$l 0.1% formic acid for each sample.

Aliquots corresponding to 2% of desalted kinase assay output were injected for mass spectrometric analysis on the Orbitrap Eclipse Tribrid system described above. The HPLC conditions were as follows: PepMap100 C18 column, 75 $\mu$m × 25 cm with 2 h RP-HPLC gradient, and 300 nl/min flow rate. The instrument method is as

described: data-dependent selection of precursor ions was performed in Cycle Time mode, with 3 s in between Master Scans, using an intensity threshold of 2e4 ion counts and applying dynamic exclusion (n = 1 scans within 30 s for an exclusion duration of 60 s and ±10 ppm mass tolerance). Monoisotopic peak determination was applied and charge states 2–6 were included for CID/ETD toggle MS/MS scans (quadrupole isolation mode; 1.6 m/z isolation window). The resulting fragments were detected in the Orbitrap at 15,000 resolutions with standard AGC target. Peptide amino acid sequence and phosphorylation site location were confirmed by manual annotation of the peptide fragment ions from MS/MS scans. In image S3E, LC-MS base peak chromatogram of in vitro-ERK2 kinase assay output (2% aliquot). Upper panel; unmodified mCx37-peptide 1, FYLPMGEGPSSPPCP (Cx37 265–279), is shown at m/z 789.8516 (z = 2); the methionine-oxidized form of this peptide is shown at m/z 797.8496 (z = 2). CID MS/MS spectrum of m/z 829.85 showing phosphosite localization at S275. Lower panel; unmodified ERK-Control Peptide, PRSARPLSPQNSPTG, is shown at m/z 522.2790 (z = 3) and phosphosphorylated Cntrl is shown at m/z 548.9340 (z = 3). ETD MS/MS spectrum of m/z 548.94 showing phosphosite localization at amino acid position 8.

### Intracellular Ca$^{2+}$ measurements

X-rhod-1 is a calcium indicator AM ester which exhibits enhanced fluorescence upon binding to free Ca$^{2+}$ (53). Cells were loaded with 1 $\mu$M X-rhod-1 AM ester diluted into culture media, incubated for 15 min at 37°C. Before imaging, cells were washed in an indicator-free medium to remove any dye that is non-specifically associated with the cell surface, and then incubated for a further 30 min to allow complete deesterification of intracellular AM esters. X-rhod-1 were imaged at >605 nm. Fluorescence intensity of the whole cell was measured for each overexpression condition and results were plotted as relative fluorescence changes to the control HUVEC as (F-Fo)/Fo, where F is the fluorescence intensity of construct-expressing cells and Fo is the fluorescence intensity of the control HUVEC. Background X-rhod-1 fluorescence was measured with HUVEC preincubated with 10 mM EGTA, and the value was subtracted from each sample.

### Statistical analysis

Data are represented and displayed as mean ± SEM and derived from three independent experiments as indicated in figure legends, if not otherwise mentioned. For quantitation of fluorescence intensity from fixed material, 200–300 cells were analyzed for each individual experiment for each condition. All statistical analyses were performed using GraphPad Prism. Unpaired two-tailed *t* tests were used to compare datasets with two groups, and a Welch's correction was applied when data normalized to the control values were being assessed. Data with three or more groups were compared using one-way ANOVA with Dunnett's post hoc test. For comparing two different independent conditions with three or more groups two-way ANOVA was applied followed by post hoc *t* test with Bonferroni correction or with Sidak's multiple comparisons test was performed.

## Data Availability

No special software or quantification code script has been generated in this study. Any further information and requests for resources and reagents should be directed to KK Hirschi (kkh4yy@virginia.edu).

## Supplementary Information

## Acknowledgements

KK Hirschi is supported by NIH (R01HL146056). MJ Humphries is supported by BBSRC, UK (BB/V001140/1) and Cancer Research UK (CRUK, DRCRPG-May21\1000002). BR Acharya, JS Fang and KK Hirschi conceived the project.

### Author Contributions

BR Acharya: conceptualization, resources, data curation, formal analysis, supervision, validation, investigation, methodology, and writing—original draft, review, and editing.
JS Fang: investigation, methodology, and writing—review and editing.
ED Jeffery: formal analysis, investigation, methodology, and writing—review and editing.
NW Chavkin: investigation and writing—review and editing.
G Genet: investigation and writing—review and editing.
H Vasavada: investigation and writing—review and editing.
EA Nelson: investigation.
GM Sheynkman: resources, investigation, and writing—review and editing.
MJ Humphries: funding acquisition, investigation, and writing—review and editing.
KK Hirschi: conceptualization, supervision, funding acquisition, and writing—review and editing.

### Conflict of Interest Statement

The authors declare that they have no conflict of interest.

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
