## [Reviewer comments · Life Science Alliance]

Life Science Alliance

Cx37 Sequestering of pERK in the Cytoplasm Promotes p27-mediated Endothelial Cell Cycle Arrest

Author information redacted

DOI: <https://doi.org/10.26508/lsa.202201685>

Corresponding author(s): Author information redacted

Review Timeline:

Submission Date:	2022-08-22
Editorial Decision:	2022-09-28
Revision Received:	2023-03-15
Editorial Decision:	2023-04-14
Revision Received:	2023-05-03
Accepted:	2023-05-04

Transaction Report:

September 28, 2022

Re: Life Science Alliance manuscript #LSA-2022-01685-T

Dr. Bipul R. Acharya
University of Manchester
Micheal Smith Building
Manchester M13 9PT
United Kingdom

Dear Dr. Acharya,

Thank you for submitting your manuscript entitled "Connexin 37 Sequestering of Activated-ERK in the Cytoplasm Promotes p27-mediated Endothelial Cell Cycle Arrest" to Life Science Alliance. The manuscript was assessed by expert reviewers, whose comments are appended to this letter. We invite you to submit a revised manuscript addressing the Reviewer comments.

Thank you for this interesting contribution to Life Science Alliance. We are looking forward to receiving your revised manuscript.

Sincerely,

B. MANUSCRIPT ORGANIZATION AND FORMATTING:

Reviewer #1 (Comments to the Authors (Required)):

In the study by Acharya et al. the authors aim to show that Connexin37 expression can be induced by arterial shear stress. The authors further aimed to show links between binding and activation of the ERK/pERK which promotes nuclear Foxo3a localization, inducing p27 expression and thereby cell cycle arrest. In general, the manuscript shows many of these aspects, but the interpretation of some of the results may not be as specific as stated by the authors. While undoubtedly changes are seen, the overall direct pathways linking these may still require further clarification. The manuscript including figures and legends would benefit from a thorough review to increase clarity throughout and correct mistakes in labeling and spelling. As the data is presented, it is quite difficult to follow the logic and flow in many parts.

General comments:

1. Throughout the manuscript, the authors state that investigations are linked with arterial gene expression. As studies were primarily performed in venous EC in vitro, this should be altered to reflect either venous genes or generally vascular EC genes were studied.
 - a. The authors state in methods that arterial EC were used but no data is provided for these
2. Cx37 is expressed in arterial EC. What is the contribution of endogenous Cx37?
 - a. Why does Cx37 expression appear to be primarily nuclear (S1B)
 - b. How do the authors define high/ low expression in these studies in HUVEC, is this compared to endogenous expression?
 - c. Authors should clarify if all cells used had prior Cx37KD by shRNA
 - d. Was gap junction function assessed
3. Initial experiments using the Cx37 mutants suggest a role for the Cx37-CT domain in the control of p27, through ERK but there is a lack of clarity on how this is regulated. In the data, Cx37-TA (non-gap junction function) forms complexes with ERK, but does not increase its phosphorylation. This suggests that Cx37 binding is only part of the process, but may require gap junction functions, or control of other proteins in complex, to phosphorylate ERK at the plasma membrane. In the discussion, it is suggested Ca²⁺ (or regulation of) may be involved. If this data exists, why is it not included?
 - a. Why is Cx37 Δ CT (no Cx37CT) increasing pERK and why is this not associated with an increase in G1 block
 - b. It is also not clear what distinction is being made between early and late G1. Are these functionally distinct points in cell cycle-regulated by specific markers?
4. There needs to be an increase in the rigor of some of the reporting in the manuscript. For example, immunofluorescence detection is used a lot throughout the manuscript as a mechanism to show specific subcellular localization and protein interaction. Given the limits of resolution of immunofluorescence, this is not appropriate in cases that are not confirmed by secondary methods such as nuclear isolation or protein-protein interaction study (co-ip, proximity ligation etc). While this is clarified for some proteins it is not done consistently throughout the manuscript.
 - a. Suggesting that both ERK and Cx37 bind microtubules based on immunofluorescent co-localization studies is insufficient (Figure 3B-C) especially when corroborating evidence in the literature is cited as a review article that does not mention Cx37
 - b. co-immunoprecipitation experiments do not show direct protein interactions, but rather complexes. Further studies would be required to call this an interaction
 - c. "unfragmented nuclei" as an indicator of cell death is not a great metric here and if required should be replaced with an appropriate cell death assay
 - d. There are some inconsistencies with the expression and imaging e.g. S1g scales appear different in p27 images and there appears to be more nuclear staining for the other CX37 mutants than controls
 - e. 1c states p27mRNA levels - what are the units here
 - f. Some control blots e.g. ERK-total are missing from some images
 - g. There is a general inconsistency in reporting what was used to control for protein loading throughout.
 - h. In flow cytometry analysis, how were gates applied? There appear to be a lot of excluded cells based on the representative images. This may affect overall percentages.
 - i. How was transfection efficiency for Fucci assessed, if they are a mixed population there may be an overestimate of early G1 states.

5. Figures need to be revised for accuracy in reporting (some examples below)
 - a. Figure 2A-C it is not clear that C is a quantification of A-B as the Cx37 tag (GFP/ HA) is different than described - why were both -HA and -GFP tags used.
 - b. Data appear to be missing, Fig S3H does not show Fox3aKD alone in the blots and the data do not match the graph between figure 3h-l as the graph appears to show Cx37 increases cMYC
 - c.
 - d. Figure 4A-F none of the data for Cx37KD is shown
 - e. S1I, are upper panels representative of CX37? In the control GFP-control or empty GFP vector signal or Cx37 antibody
 - f. S1K is referred to as KD+GFP tagged - this is presumable Cx37-GFP
 - g. S2E is referred to as CX37 O/E, this is presumably Cx37-GFP transfected
 - h. Check molecular weight markers in 2a-b for 75kda, they are at different levels compared to the same protein.
6. There are quite a lot of issues with labeling throughout that make it difficult to access the data properly some examples below -
 - a. the text requires to be checked thoroughly
 - b. Figure legends do not currently accurately report what is shown (also not found in results) - this makes some of the data uninterpretable
 - c. The word control is used in many different forms throughout
 - d. There is a lack of clarity on some experiments for example use of DAPT and DOX. The purpose of these experiments is not stated.
 - e. Mislabeled figures e.g. "SH-I" should be S2H-I
 - f. Spelling errors in some images e.g. "nuclues"

Reviewer #2 (Comments to the Authors (Required)):

In this manuscript, the authors aimed to investigate how the induced expression of Cx37 (gap junction protein) upregulates the cyclin-dependent kinase inhibitor p27 to enable the endothelial growth suppression and arterial specification. They expressed the wild type and regulatory domain mutants of Cx37 in cultured endothelial cells with the Fucci cell cycle reporter. The results showed that the Cx37/pERK/Foxo3a/p27 signaling axis functions downstream of arterial shear stress to promote endothelial late G1 state and enable upregulation of arterial genes. They did a lot of work. In general, the methods used in this study are feasible and the data are solid to support the conclusion. It could be accepted for publication.

Please add the line numbers in the manuscript.

In the figure legends, explain what the arrowheads indicate?

In the second paragraph of the introduction, where is CX37 present in vascular ECs and how is it transported in the cytoplasm?

In Fig 4 legend, E) and F) FSS and pERK protein level, and effect of Cx37 KD. F) Effect of Foxo3a KD and Kinase inhibitors on FSS induced HUVEC-FastFucci cell cycle.

In my opinion, people usually study CX37 and CX43 together. It is better to discuss CX43 in Discussion.

RESPONSES TO REVIEWERS (Point-by-Point)**Reviewer #1:**

In the study by Acharya et al. the authors aim to show that Connexin37 expression can be induced by arterial sheer stress. The authors further aimed to show links between binding and activation of the ERK/pERK which promotes nuclear Foxo3a localization, inducing p27 expression and thereby cell cycle arrest. In general, the manuscript shows many of these aspects, but the interpretation of some of the results may not be as specific as stated by the authors. While undoubtedly changes are seen, the overall direct pathways linking these may still require further clarification. The manuscript including figures and legends would benefit from a thorough review to increase clarity throughout and correct mistakes in labelling and spelling. As the data is presented, it is quite difficult to follow the logic and flow in many parts.

Response: We thank the reviewer for the constructive feedback. We have addressed each comment in our revised manuscript, as detailed below.

1. Throughout the manuscript, the authors state that investigations are linked with arterial gene expression. As studies were primarily performed in venous EC in vitro, this should be altered to reflect either venous genes or generally vascular EC genes were studied.

a. The authors state in methods that arterial EC were used but no data is provided for these

Response: As pointed out by the reviewer, we focus our investigations on arterial endothelial gene expression, as Cx37 function has been linked to arterial endothelial gene expression in previous publications from our group and others. We agree with this comment in that the effects of Cx37 function on venous gene expression is an important factor in understanding overall arterial-venous specification. Therefore, **we quantified the mRNA expression by qPCR for venous genes in HUVEC** with endogenous Cx37 knockdown and overexpression of wild type Cx37 and various mutant constructs following exposure to arterial fluid shear stress (FSS). Venous genes did not change (**Ephb4 and Nr2f2; revised Fig. S6C**), which is in contrast to our findings on arterial gene (Hey2, Efnb2, GJA5) expression that we presented in our initial submission (Fig. 4H). We have now included our new data on venous gene expression in our revised manuscript.

In parallel to HUVEC, **we also used Human Aortic Endothelial Cells (HAEC)** to determine whether arterial FSS can trigger Cx37/pERK/Foxo3a/p27 signalling pathway to induce late G1 cell cycle arrest in these cells (**revised Fig. S6D, E**). Indeed, arterial FSS induced late G1 arrest in HAEC, and late G1 arrest following arterial FSS was abrogated if Notch signalling and ERK phosphorylation were inhibited, or if Cx37, Foxo3a or p27 were knocked down in HAEC (**Fig. revised S6E**). Also, arterial FSS induced arterial genes, but not venous gene expression in HAEC (**Fig. revised S6F**). These data are now included in our revised manuscript. Hence, we kept our statements that arterial FSS only induces “arterial-specific gene” expression in endothelial cells.

2. Cx37 is expressed in arterial EC. What is the contribution of endogenous Cx37?

Response: Cells that are not expressing exogenous Cx37 constructs do not activate the signalling cascade suggesting that a threshold level of Cx37 expression is required for p27 upregulation and late G1 arrest, which is not achieved by only endogenously expressed Cx37 in HUVEC. The overexpression of Cx37 constructs is at least 2-fold higher than endogenous Cx37 (please see attached immunoblot); however, endogenous Cx37 expression was not altered with the exogenous expression of any Cx37 construct.

a. Why does Cx37 expression appear to be primarily nuclear (S1B)

Response: Figure S1B is showing the immunostaining of Cx37 using Abcam Rabbit-polyclonal antibodies, and we agree with reviewer that there is substantial nuclear signal with this antibody. Unfortunately, lack of very high-fidelity Cx37 antibodies for immunostaining in HUVEC is a known problem in the field, and we found the same problem after testing several commercially available antibodies. Often, we found non-specific staining throughout and reduced or no punctate staining at junction/membrane. With the antibody we employ in our manuscript, we could detect junctional punctate staining that is greatly reduced with knockdown of endogenous Cx37. The background nuclear staining with this antibody is also visible with knockdown, suggesting that this nuclear signal is due to non-specific epitope binding by the rabbit polyclonal Cx37 antibodies.

Because of this issue, we used an anti-GFP antibodies to track overexpressed Cx37 in HUVEC, and we did not observe any nuclear staining of Cx37 (Fig. 1A). This further supports that the nuclear signal observed with the anti-Cx37 antibodies is non-specific.

b. How do the authors define high/ low expression in these studies in HUVEC, is this compared to endogenous expression?

Response: We understand this may be confusing, so we have replaced low and high expression of Cx37 with “**Steady State**”, defined as endogenous expression in control, static culture conditions, and “**Elevated Cx37 Expression**”, defined as increased expression with exogenous expression of Cx37 constructs or in response to arterial FSS (Fang, et. al; *Nature Commun.* 2017), **in revised text and revised graphical abstract (Summary blurb now).**

c. Authors should clarify if all cells used had prior Cx37KD by shRNA

Response: Thank you for this suggestion. We have now clearly stated in the text whether the cells are overexpressing Cx37 constructs in the presence of endogenous Cx37 or in a Cx37-knockdown background. This is now included in the **revised Results section.**

d. Was gap junction function assessed?

Response: This is an important question that we agree should be addressed. To investigate gap junction function, we performed the Scrape/loading Lucifer yellow dye transfer assay in HUVECs expressing endogenous Cx37 and the different Cx37 exogenous constructs and measured **Gap Junction Intercellular Communication (GJIC)**. We found that Cx37FL and Cx37 Δ CT moderately increased GJIC above endogenous/background levels, but other constructs failed to do so. However, none of these exogenous constructs dampened GJIC below control levels (**revised Fig. S2C**), suggesting their expression did not affect basal GJIC. Nevertheless, we found that increased GJIC, albeit modestly with Cx37FL and Cx37 Δ CT, is important for Ca²⁺ abundance and necessary for ERK activation in the cells following Cx37 exogenous expression, so, functional gap junction is indispensable this signalling pathways. We have now incorporated these data in the revised manuscript (**revised Fig. S2B, C**).

3. Initial experiments using the Cx37 mutants suggest a role for the Cx37-CT domain in the control of p27, through ERK but there is a lack of clarity on how this is regulated. In the data, Cx37-TA (non-gap junction function) forms complexes with ERK, but does not increase its phosphorylation. This suggests that Cx37 binding is only part of the process, but may require gap junction functions, or control of other proteins in complex, to phosphorylate ERK at the plasma membrane. In the discussion, it is suggested Ca²⁺ (or regulation of) may be involved. If this data exists, why is it not included?

Response: We thank reviewer for raising this question. Our results support that the c-terminal domain of Cx37 is responsible for pERK interaction in association with Microtubule (MT), which sequesters pERK in the cytoplasm. Because we showed that Cx37 Δ CT lacked the ability to pulldown pERK in co-IP analysis, and that MT depolymerization by nocodazole reduced pERK interaction with Cx37, we suggest a co-dependency among Cx37-c-terminal domain, pERK and MT. We have now presented more experimental evidence in support of our hypothesis (**revised Figure S4**). We now show Cx37FL-GFP co-immunoprecipitates MT (**blotted for alpha-tubulin; revised Fig. S4F**), Cx37TA and CX37Tail could similarly pulldown MT, but Cx37 Δ CT failed to do so, indicating, Cx37 tail domain is interacting with MT **revised Fig. S4F**) and this is important for pERK cytoplasmic sequestration. To increase confidence, and as suggested by reviewer, we performed the proximity ligation assay (PLA) between MT and Cx37 (**revised Fig. S4G, S5A**). As known, PLA only shows protein-protein interactions and provides PLA dots when two proteins are within <40 nm distance; and possibly participate in a multimeric-complex. Unlike other constructs, only HA- Cx37 Δ CT failed to increase PLA dots/cells when expressed exogenously in Cx37-MT PLA assays, which further suggests that Cx37 and MT interact and their interaction is mediated by the Cx37 c-terminal tail. Also, we performed Cx37 and pERK PLA assays (**revised Fig. S5B, C**). Enhanced PLA dots/cells was observed when Cx37FL was expressed but, as soon as the MT is depolymerized, the PLA association is abolished, suggesting that Cx37, pERK and MT build a tripartite interaction when Cx37 is elevated above steady state levels in the cell.

As the reviewer suggests, we also found that this tripartite interactions among Cx37, MT and pERK are not only the part of this process, rather the big “unknown” in this mechanism was how Cx37 overexpression induces ERK phosphorylation. Our Scrape/loading GJIC functional assay data, which are now included in our revised manuscript, showed that overexpression of either Cx37 or Cx37 Δ CT increases GJIC, and we also have now shown that both of them increase intracellular free calcium (**revised Fig. S4B, C**). We measured the intracellular Ca²⁺ following expression of the HA-Cx37 construct using the Ca²⁺ indicator, X-Rhod-1 AM-Ester, which exhibits enhanced fluorescence after binding to free Ca²⁺. It binds specifically to Ca²⁺ with high affinity. Both Cx37FL and Cx37 Δ CT showed enhanced fluorescence compared to other constructs, suggesting their expression induces elevated intracellular Ca²⁺. Interestingly, Ca²⁺ is an upstream regulator and acts via the Raf/MEK/ERK pathway to promote ERK activation. Indeed, when we chelated intracellular Ca²⁺ with BAPTA-AM, it inhibited ERK activation following Cx37FL expression, indicating GJIC is an important for Ca²⁺ elevation in HUVEC to facilitate this signalling pathway. Although the source of free Ca²⁺ is not confirmed in our studies (influx through GJs or release from cell-interior ER membrane), increased abundance of Ca²⁺ in Cx37-overexpressing cells seems to be another important part of this mechanism.

a. Why is Cx37 Δ CT (no Cx37CT) increasing pERK and why is this not associated with an increase in G1 block

Response: We show that Cx37 c-terminal interaction with pERK is essential for its cytosolic sequestration. Cx37 Δ CT could induce ERK phosphorylation by increasing intracellular Ca²⁺, but Cx37 Δ CT failed to sequester pERK in the cytosol due to the lack of the Cx37 c-terminal domain that mediates the tripartite interaction among MT, Cx37 and pERK. Subsequently, pERK travels to the nucleus and phosphorylates Foxo3a, promoting Foxo3a efflux to the cytoplasm for degradation. This prevents p27 upregulation and allows the cell to progress past G1 phase and into S phase.

b. It is also not clear what distinction is being made between early and late G1. Are these functionally distinct points in cell cycle-regulated by specific markers?

Response: Indeed, early G1 and late G1 are two distinct cell cycle states distinguishable using the FUCCI reporter system. These states, specifically in HUVECs, have been well characterized in a previous publication by our group (Chavkin et al. *Nature Commun.* 2022) as well as others in other cell types. In our previous publication, we extensively described the proteomic, transcriptional, and chromatin availability variations in Fucci-expressing HUVEC in early G1 and late G1. Our findings are consistent with other publications that define early G1 and late G1 as distinct cell cycle states with different molecular profiles. We also included a movie of the same HUVEC-FUCCI cells undergoing several cell divisions over 72 hours where the distinct cell cycle states are clearly visible (early G1 as blank, late G1 as red, and S/G2/M as green) while the cells passed through the cell cycle. For your reference, below, we included papers by others that use the FUCCI reporter system in vitro to characterize distinct cell cycle states.

1. Singh AM, Trost R, Boward B, Dalton S. **Utilizing FUCCI reporters to understand pluripotent stem cell biology**. *Methods*. 2016 May 15;101:4-10. doi: 10.1016/j.ymeth.2015.09.020. Epub 2015 Sep 21. PMID: 26404921; PMCID: PMC4801677.

This is a methods paper describing how the Fucci reporter system can be used to study stem cell biology by separating cells into various cell cycle states (Early G1, Late G1, S/G2/M).

2. Singh AM, Chappell J, Trost R, Lin L, Wang T, Tang J, Matlock BK, Weller KP, Wu H, Zhao S, Jin P, Dalton S. **Cell-cycle control of developmentally regulated transcription factors accounts for heterogeneity in human pluripotent cells**. *Stem Cell Reports*. 2013 Dec 5;1(6):532-44. doi: 10.1016/j.stemcr.2013.10.009. Erratum in: *Stem Cell Reports*. 2014 Mar 11;2(3):398. PMID: 24371808; PMCID: PMC3871385.

This study showed that hESCs have varying expression of developmental transcription factors in various cell cycle states.

3. Jang J, Han D, Golkaram M, Audouard M, Liu G, Bridges D, Hellander S, Chialastri A, Dey SS, Petzold LR, Kosik KS. **Control over single-cell distribution of G1 lengths by WNT governs pluripotency**. *PLoS Biol*. 2019 Sep 26;17(9):e3000453. doi: 10.1371/journal.pbio.3000453. PMID: 31557150; PMCID: PMC6782112.

This study uses the Fucci reporter to investigate the role of WNT signaling on length of time in different cell cycle states.

4. Shimono H, Kaida A, Homma H, Nojima H, Onozato Y, Harada H, Miura M. **Fluctuation in radioresponse of HeLa cells during the cell cycle evaluated based on micronucleus frequency**. *Sci Rep*. 2020 Nov 30;10(1):20873. doi: 10.1038/s41598-020-77969-0. PMID: 33257719; PMCID: PMC7705701.

This study uses Fucci-expressing HeLa cells to determine the differential response to irradiation in various cell cycle states.

5. Aureille J, Buffière-Ribot V, Harvey BE, Boyault C, Pernet L, Andersen T, Bacola G, Balland M, Fraboulet S, Van Landeghem L, Guilluy C. **Nuclear envelope deformation controls cell cycle progression in response to mechanical force**. *EMBO Rep*. 2019 Sep;20(9):e48084. doi: 10.15252/embr.201948084. Epub 2019 Aug 1. PMID: 31368207; PMCID: PMC6726894.

This study investigates the functional role of mechanical force on nuclear envelope deformation and cell cycle progression using the Fucci reporter to determine length of cell cycle states.

4. There needs to be an increase in the rigor of some of the reporting in the manuscript. For example, immunofluorescence detection is used a lot throughout the manuscript as a mechanism to show specific subcellular localization and protein interaction. Given the limits of resolution of immunofluorescence, this is not appropriate in cases that are not confirmed by secondary methods such as nuclear isolation or protein-protein interaction study (co-ip, proximity ligation etc). While this is clarified for some proteins it is not done consistently throughout the manuscript.

a. Suggesting that both ERK and Cx37 bind microtubules based on immunofluorescent co-

localization studies is insufficient (Figure 3B-C) especially when corroborating evidence in the literature is cited as a review article that does not mention Cx37

Response: We appreciate the reviewer's suggestion. As explained in a previous response, to increase the rigor of our experimental findings that suggest pERK builds a complex with Cx37 c-terminal domain and MT, we now include a co-immunoprecipitation analysis of Cx37 and MT. We found that, in **revised Fig. S4F**, Cx37FL, TA and Tail domain co-immunoprecipitated MT in a Cx37-GFPtrap immunoprecipitation experiment. However, Cx37 Δ CT was unable to pulldown MT, suggesting the c-terminal tail is the interaction zone for Cx37 with MT. In addition, we have performed a Rabbit/Mouse Proximity Ligation Analysis (PLA) with MT (mouse) and Cx37 (Rabbit) in Cx37-expressing HUVEC. We found more PLA dots/cell in Cx37FL cells (**revised Fig. S4G, and S5A**) compared to the empty vector control and Cx37 Δ CT cells, consistent with our previous observation that the Cx37 c-terminal tail is an interaction site for MT. We have also showed that PLA dots/cells for pERK interaction with Cx37FL is almost completely abolished after Nocodazole treatment to depolymerize MT (**revised Fig. S5A, B**). These results suggest that the Cx37 c-terminal tail and ERK undergo a tripartite interaction complexed within <40 nm distance (by PLA results) in the cytoplasm, and MT facilitates this interaction.

b. co-immunoprecipitation experiments do not show direct protein interactions, but rather complexes. Further studies would be required to call this an interaction

Response: We agree with this point and replaced "binding" with "**interaction**" in our revised manuscript. In addition, we performed additional analysis to supplement our co-immunoprecipitation studies that suggest that Cx37 is part of a MT-dependent multimeric protein complex which helps to sequester pERK in the cytoplasm. We performed PLA analysis with Cx37-MT and Cx37-pERK to show similar interactions, which suggests this interaction, as a complex, is occurring within <40 nm distance. We have explained these data in previous responses, and we have now added these data to the manuscript as **Fig. S4G, S5A-S5C**.

c. "unfragmented nuclei" as an indicator of cell death is not a great metric here and if required should be replaced with an appropriate cell death assay

Response: We agree with this point and, therefore, have now included a MTT (a yellow tetrazole that is reduced to purple formazan by mitochondrial reductase in living cells) based cell viability assay, which is a colorimetric assay for assessing cell metabolic activity for checking the viability of the cells. We performed the MTT assay on HUVEC expressing different Cx37 constructs after 96h of transduction. We did not observe any reduction in viability in Cx37-expressing HUVECs in comparison to controls (**revised Fig. S2A**).

d. There are some inconsistencies with the expression and imaging e.g. S1g scales appear different in p27 images and there appears to be more nuclear staining for the other CX37 mutants than controls

Response: We think that reviewer is referring to **S1I, not S1g**. If so, we understand the confusion. The control image in figure S1I (**which is now S1J**) shows the Cx37KD HUVEC, where p27 nuclear intensity is reduced due to Cx37 knockdown, which is then rescued when Cx37FL is re-expressed in KD cells but could not be rescued by the other exogenous Cx37 constructs. So, the control in these experiments is the Cx37KD cells not the control-shRNA expressing cells. To mitigate this confusion, now we have included a p27 image expressing control-shRNA.

e. 1c states p27mRNA levels - what are the units here

Response: We thank the reviewer for pointing out this. As this is a relative measurement, the units should be fold change vs. control. We have now updated it in the figure (**revised Fig. 1C**).

f. Some control blots e.g., ERK-total are missing from some images

Response: The total ERK blotting for **revised Fig 3E, 3I, S4D, S5D and S5F** are now included.

g. There is a general inconsistency in reporting what was used to control for protein loading throughout.

Response: The total protein of the cell lysates was quantified with BCA assay for Western blots and IP experiments. All immunoblots include a loading control to ensure uniformity of protein lysate loading for each assay. For whole cell lysates, we used beta-actin as a loading control but, for nuclear versus cytosolic fraction experiments, we could not use beta-actin as a loading control rather we used Lamin A and Vimentin which are the bonafide nuclear and cytosolic marker. For GFP-trap immunoprecipitation assay, the protein concentration of every cell lysate was measured and normalized with the control (GFP-empty vector expressing cells) before loading with GFP-trapping beads, the equal volume of final elutes were loaded for immunoblotting. As the protein expression level of every GFP-Cx37 constructs were comparable, we think the abundance of every Cx37-GFP protein trapped with GFP-beads should be equivalent.

h. In flow cytometry analysis, how were gates applied? There appear to be a lot of excluded cells based on the representative images. This may affect overall percentages.

Response: Gating was applied uniformly to all the samples. Flow cytometry gates were applied as described in previous publications using the FUCCI reporter system *in vitro* (Chavkin *et al*, *Nature Commun.* 2022). This previous paper also validated the specific cell cycle states in these gates using known markers of early G1 and late G1, as well as functional DNA synthesis and cell cycle-related protein phosphorylation. Technical limitations of flow cytometry and the FUCCI reporter system yield data outside of the gates (cell debris, auto-fluorescence, etc.), but these data are not included when calculating the overall percentages and therefore do not affect the results presented in this manuscript.

i. How was transfection efficiency for FUCCI assessed, if they are a mixed population there may be an overestimate of early G1 states.

Response: This is a stable cell line and kept under selection with puromycin antibiotics. In addition, the expression is regularly checked by flow cytometric analysis to confirm that there is not a reduction in FUCCI fluorescence with subsequent passages.

5. Figures need to be revised for accuracy in reporting (some examples below)

a. Figure 2A-C it is not clear that C is a quantification of A-B as the Cx37 tag (GFP/ HA) is different than described - why were both -HA and -GFP tags used.

Response: We appreciate this suggestion and have now checked the manuscript thoroughly for such mistakes. HA-Cx37 labelling in **revised Figure 2B and 2K** is now replaced with Cx37-GFP. HA-Cx37 was used in Foxo3a rescue studies, performed in the Cx37-GFP cells. The HA-tagged version of Cx37 was required for expression in Fucci cells as Fucci itself has green fluorescence.

b. Data appear to be missing, Fig S3H does not show Fox3aKD alone in the blots and the data do not match the graph between figure 3h-I as the graph appears to show Cx37 increases cMYC

Response: We thank the reviewer for pointing this out. We have replaced the old immunoblot with a new one which includes the Foxo3aKD in **revised Figure S3H**, and the quantitation graph in **revised Figure S3I** labels are now corrected, accordingly.

d. Figure 4A-F none of the data for Cx37KD is shown

Response: We have now shown the pannel of the effect of Cx37KD on relative nuclear vs. cytoplasmic fluorescence intensity of Foxo3a (Fig. 4B) and pERK (Fig. 4D), and the relative phosphorylation level of ERK (Fig. 4E, F).

e. S1I, are upper panels representative of CX37? In the control GFP-control or empty GFP vector signal or Cx37 antibody

Response: We apologize for this unintentional confusion. The “control” in the upper panel of figure S1I (now **revised Figure S1J**) is the empty-GFP vector, and the antibodies that stained that panel are anti-GFP antibodies. We have now corrected this for better understanding.

f. S1K is referred to as KD+GFP tagged - this is presumable Cx37-GFP

Response: We thank the reviewer for the suggestion; the label is now changed to KD+Cx37-GFP in **revised Figure S1L**.

g. S2E is referred to as CX37 O/E, this is presumably Cx37-GFP transfected

h. Check molecular weight markers in 2a-b for 75kda, they are at different levels compared to the same protein.

Response: We thank the reviewer for the suggestion; the label is now changed to “Cx37-GFP+” in revised **Figure S3E**.

We understand that the position of molecular weight (MW) marker 75 is different in figure 2a for Foxo3a than in figure 2B for Foxo1 and Foxo4. The predicted MW of Foxo1 is ~70kDa but it always exhibits a band between 75 to 80 kDa. Although Foxo3a has a predicted MW of ~71 kDa, the observed MW is about 72 – 97 kDa, and we always find that it runs at around 80-85 kDa. The predicted MW of Foxo4 is 65 kDa and it runs around the same size. This is the reason why we should see the MW marker 75 at different places in figure 2A and 2B.

6. There are quite a lot of issues with labeling throughout that make it difficult to access the data properly some examples below -

a. the text requires to be checked thoroughly

Response: We appreciate the reviewer’s suggestion; the text has been thoroughly checked and rewritten to correct errors.

b. Figure legends do not currently accurately report what is shown (also not found in results) - this makes some of the data uninterpretable

Response: We have now revised the Figure legends, so that they describe the figures better.

c. The word control is used in many different forms throughout

Response: To clarify, we have now defined the “Control” in respective places throughout the text.

d. There is a lack of clarity on some experiments for example use of DAPT and DOX. The purpose of these experiments is not stated.

Response: We have now included the respective use of DAPT as a Notch inhibitor and DOX for induced knockdown of Cx37 in the revised **Results** section.

e. Mislabeled figures e.g. "SH-I" should be S2H-I

Response: We have corrected this error in the revised manuscript, it is now revised **S3H-I**.

f. Spelling errors in some images e.g. "nuclues"

Response: We have now corrected this error in revised **Fig. 3E**.

Reviewer #2:

In this manuscript, the authors aimed to investigate how the induced expression of Cx37 (gap junction protein) upregulates the cyclin-dependent kinase inhibitor p27 to enable the endothelial growth suppression and arterial specification. They expressed the wild type and regulatory domain mutants of Cx37 in cultured endothelial cells with the Fucci cell cycle reporter. The results showed that the Cx37/pERK/Foxo3a/p27 signaling axis functions downstream of arterial shear stress to promote endothelial late G1 state and enable upregulation of arterial genes. They did a lot of work. In general, the methods used in this study are feasible and the data are solid to support the conclusion. It could be accepted for publication.

Response: We thank the reviewer for his/her encouraging words and supportive feedback on our manuscript.

Please add the line numbers in the manuscript.

Response: We have now added line numbers in the manuscript.

In the figure legends, explain what the arrowheads indicate?

Response: We have now explained what the arrowheads are indicating in the revised figure legends.

In the second paragraph of the introduction, where is CX37 present in vascular ECs and how is it transported in the cytoplasm?

Response: We have now added discussion on Cx37 localization and its cytosolic transport in the **revised Introduction section**.

In Fig 4 legend, E) and F) FSS and pERK protein level, and effect of Cx37 KD. F) Effect of Foxo3a KD and Kinase inhibitors on FSS induced HUVEC-FastFUCCI cell cycle.

In my opinion, people usually study CX37 and CX43 together. It is better to discuss CX43 in Discussion.

Response: We have now added discussion on Cx43 and pERK in the **revised Discussion section**.

April 14, 2023

RE: Life Science Alliance Manuscript #LSA-2022-01685-TR

Dr. Bipul R. Acharya
University of Manchester
Micheal Smith Building
Manchester M13 9PT
United Kingdom

Dear Dr. Acharya,

Thank you for submitting your revised manuscript entitled "Cx37 Sequestering of pERK in the Cytoplasm Promotes p27-mediated Endothelial Cell Cycle Arrest". We would be happy to publish your paper in Life Science Alliance pending final revisions necessary to meet our formatting guidelines.

- please address Reviewer 1's remaining comments
- please add ORCID ID for secondary corresponding author-they should have received instructions on how to do so
- please add a category for your manuscript to our system
- please upload your graphical abstract as a separate file, labeled as graphical abstract
- please use the [10 author names, et al.] format in your references (i.e. limit the author names to the first 10)

A. FINAL FILES:

B. MANUSCRIPT ORGANIZATION AND FORMATTING:

Sincerely,

Reviewer #1 (Comments to the Authors (Required)):

The revisions of the manuscript by Acharya et al. have addressed my concerns from the first review. A few other points should be considered.

- FSS gene expression on control cells should be shown in Figure 4H as it appears that Cx37KD of endogenous Cx37 reduces differential gene expression changes to FSS, when comparing to data in 4i
- o For clarity in 4H, it would be good to include in the figure labels that Cx37FL and Delta T transfections are in addition to Cx37KD.
- A clarifying statement should be included to highlight that nuclear staining for Cx37 is non-specific, as stated in the reviewer's response.
- Clarify, in the text, which genes relate to venous and arterial (line 220)

The manuscript still requires to be read very thoroughly for errors in reporting - check throughout - some examples below.

- Figure s1b images for DAPT stains are not shown for data quantified in c-D,
- Line 156 states Fig S3K,L - L is not present in the image
- Line 104 states figure S1i shows p21 and p57 are not upregulated by Cx37 - this is shown in figure S1G
- It is not clear why MT PLA data is split between S4G and S5A
- Graph labels e.g., Figure S2A reads "Rrelative" Please check all figure labels and legends

Reviewer #2 (Comments to the Authors (Required)):

No more comments

RESPONSES TO REVIEWERS (Point-by-Point)

Reviewer #1:

The revisions of the manuscript by Acharya et al. have addressed my concerns from the first review. A few other points should be considered.

We thank reviewer 1 for being supportive and constructive throughout, it has significantly enhanced our manuscript from its initial versions. In this version, we have endeavored to answer or rectify errors in the manuscript as suggested by reviewer 1.

- FSS gene expression on control cells should be shown in Figure 4H as it appears that Cx37KD of endogenous Cx37 reduces differential gene expression changes to FSS, when comparing to data in 4i
 - o For clarity in 4H, it would be good to include in the figure labels that Cx37FL and Delta T transfections are in addition to Cx37KD.

We thank reviewer for pointing this and we acknowledge that the control FSS data in 4H, S6C and S6F would really be helpful for evaluating the clarity of our data. Hence, we have included the control FSS panel in each of these graphs now.

- A clarifying statement should be included to highlight that nuclear staining for Cx37 is non-specific, as stated in the reviewer's response.

We have now included (and highlighted) a line in the text about the non-specific nuclear staining by Cx37 antibody, please see line 113.

- Clarify, in the text, which genes relate to venous and arterial (line 220)

We have now clarified the arterial genes and venous genes in the text respectively, please see line 228.

The manuscript still requires to be read very thoroughly for errors in reporting - check throughout - some examples below.

We thank reviewer for this caution and have checked and fixed all errors in manuscript.

- Figure S1b images for DAPT stains are not shown for data quantified in c-D,

The inhibition of FSS induced Cx37 upregulation in DAPT treated HUVEC reconfirms that the Notch activation is important for this upregulation, which we have shown and validated in Feng et. al.; Nat. Comm. 2017. Therefore, we think addition of this particular image is redundant and unnecessary for this manuscript, hence, we have just shown the quantitation in figure S1B.

- Line 156 states Fig S3K,L - L is not present in the image

We have now corrected that in the text please see line 164 (now).

- Line 104 states figure S1i shows p21 and p57 are not upregulated by Cx37 - this is shown in figure S1G

We have now corrected this in the text. Thanks you.

- It is not clear why MT PLA data is split between S4G and S5A

We agree with reviewer on this point and we have now include S5A in figure S4 as S4H.

- Graph labels e.g., Figure S2A reads "Rrelative" Please check all figure labels and legends

We thanks reviewer for pointing this and it is now corrected that in revised figure S2A.

Reviewer #2:

No more comments

May 4, 2023

RE: Life Science Alliance Manuscript #LSA-2022-01685-TRR

Dr. Bipul R. Acharya
University of Manchester
Micheal Smith Building
Manchester M13 9PT
United Kingdom

Dear Dr. Acharya,

Thank you for submitting your Research Article entitled "Cx37 Sequestering of pERK in the Cytoplasm Promotes p27-mediated Endothelial Cell Cycle Arrest". It is a pleasure to let you know that your manuscript is now accepted for publication in Life Science Alliance. Congratulations on this interesting work.

*****IMPORTANT:** If you will be unreachable at any time, please provide us with the email address of an alternate author. Failure to respond to routine queries may lead to unavoidable delays in publication.*******

DISTRIBUTION OF MATERIALS:

Again, congratulations on a very nice paper. I hope you found the review process to be constructive and are pleased with how the manuscript was handled editorially. We look forward to future exciting submissions from your lab.

Sincerely,
